# Scaffold-mediated gating of Cdc42 signalling flux

Péter Rapali[1], Romain Mitteau[1], Craig Braun[2], Aurèlie Massoni-Laporte[1], Caner Ünlü[1], Laure Bataille[1], Floriane Saint Arramon[1], Steven P Gygi[2], Derek McCusker[1]*

[1]University of Bordeaux, CNRS, European Institute of Chemistry and Biology, IBGC, UMR 5095, Pessac, France; [2]Department of Cell Biology, Harvard Medical School, Boston, United States

**Abstract** Scaffold proteins modulate signalling pathway activity spatially and temporally. In budding yeast, the scaffold Bem1 contributes to polarity axis establishment by regulating the GTPase Cdc42. Although different models have been proposed for Bem1 function, there is little direct evidence for an underlying mechanism. Here, we find that Bem1 directly augments the guanine exchange factor (GEF) activity of Cdc24. Bem1 also increases GEF phosphorylation by the p21-activated kinase (PAK), Cla4. Phosphorylation abrogates the scaffold-dependent stimulation of GEF activity, rendering Cdc24 insensitive to additional Bem1. Thus, Bem1 stimulates GEF activity in a reversible fashion, contributing to signalling flux through Cdc42. The contribution of Bem1 to GTPase dynamics was borne-out by in vivo imaging: active Cdc42 was enriched at the cell pole in hypophosphorylated cdc24 mutants, while hyperphosphorylated cdc24 mutants that were resistant to scaffold stimulation displayed a deficit in active Cdc42 at the pole. These findings illustrate the self-regulatory properties that scaffold proteins confer on signalling pathways.

*For correspondence: mccusker@iecb.u-bordeaux.fr

Competing interests: The authors declare that no competing interests exist.

## Introduction

Scaffold proteins are multivalent binding proteins that assemble and regulate the fidelity of signal transduction cascades, conferring properties ranging from signal amplification, specificity, localisation and switch-like behaviour (*Langeberg and Scott, 2015*). However, the mechanisms by which scaffolds control signalling pathways spatially and temporally are not well understood.

The budding yeast *Saccharomyces cerevisiae* provides a useful model for understanding the spatio-temporal control of scaffold signalling, since the scaffold protein Bem1 plays an important, yet incompletely defined role in the Cdc42-dependent establishment of polarised growth that the organism displays during the cell cycle. Early in the cell cycle, Cdc42 is activated at a specific site on the plasma membrane by the GEF Cdc24. Site-specific activation of Cdc42 involves landmark proteins and the scaffold Bem1, which binds Cdc24 and Cdc42 GTP (*Peterson et al., 1994*; *Zheng et al., 1995*; *Bender et al., 1996*; *Bose et al., 2001*; *Butty et al., 2002*; *Yamaguchi et al., 2007*). The interaction of the scaffold with the active GTPase and its upstream GEF led to the idea that Bem1 might constitute part of a positive feedback loop: activated Cdc42 could bind Bem1, recruiting the GEF to activate additional Cdc42, thus building up a local pool of the active GTPase (*Butty et al., 2002*; *Johnson et al., 2011*). This autocatalytic positive feedback loop underpins numerous mathematical models that have been developed to understand the mechanisms that establish polarity (*Goryachev and Pokhilko, 2006*, *2008*; *Howell et al., 2012*; *Savage et al., 2012*; *Jose et al., 2013*).

The scaffold Bem1 also interacts with the PAK Cla4, which phosphorylates Cdc24 on multiple sites (*Gulli et al., 2000*; *Bose et al., 2001*; *Wai et al., 2009*). While an initial study found that

phosphorylation of Cdc24 may reduce its interaction with Bem1 (*Gulli et al., 2000*), a parallel study did not reach the same conclusion (*Bose et al., 2001*). Phosphorylation of Rho-family GEFs in *Ustilago maydis* and *Schizosaccharomyces pombe* regulates their function by inducing GEF degradation or sequestration by 14-3-3 proteins, respectively (*Frieser et al., 2011*; *Das et al., 2015*). However, the function of Cdc24 phosphorylation in budding yeast was elusive, since the mutation of multiple phosphorylation sites did not generate an obvious phenotype in vivo (*Wai et al., 2009*). Recently, biochemical analysis reported that phosphorylation of Cdc24 reduced its GEF activity, providing a source of negative feedback during Cdc42-mediated polarisation, although the underlying mechanism is unknown (*Kuo et al., 2014*). Thus, Bem1 has been proposed to play roles in positive and negative feedback that regulate Cdc42 activity. However, it is not presently understood how Bem1 mediates these effects. Indeed, while previous studies did not find a role for Bem1 in Cdc24 GEF activity (*Zheng et al., 1995*), a recent report using a biosensor of Cdc42 activity in vivo proposed that Bem1 may boost Cdc24 GEF activity rather than acting as part of the autocatalytic positive feedback loop. It is presently unknown whether this scaffold effect on GEF activity is direct, or how this effect might contribute to Cdc42 dynamics (*Smith et al., 2013*).

Here, we reconstitute Cdc42 activity to directly test the role of the scaffold Bem1 in Cdc42 regulation. Our results suggest that the scaffold Bem1 gates signalling flux through Cdc42 by activating GEF activity in a reversible manner. This effect of the scaffold impinges upon active Cdc42 levels at the cell pole and the global organisation of cellular polarity.

## Results

### Bem1 directly increases the rate of Cdc24 GEF activity towards Cdc42

Cdc42 activity was reconstituted by purifying the GTPase from the yeast *Pichia pastoris* (*Figure 1—figure supplement 1A*, left panel), while Bem1 and Cdc24 were purified from bacteria as full-length proteins (*Figure 1—figure supplement 1A*, right panel). Working towards the goal of reconstituting Cdc42 activation, we next assessed the capacity of the eukaryotic-expressed Cdc42 to bind fluorescent mant-GDP and mant-GTP. Mant-GDP (*Figure 1—figure supplement 1B*) or mant-GTP (*Figure 1—figure supplement 1C*) were incubated with increasing concentrations of purified nucleotide-free Cdc42, and FRET between the mant moiety and Cdc42 was measured (*Figure 1—figure supplement 1B and C* insets). By fitting a simple binding model to the maximal fluorescence intensity changes over a range of Cdc42 concentrations, estimated $K_d$ of 1 µM and 0.5 µM were obtained for mant-GDP and mant-GTP, respectively, in accordance with previous reports (*Figure 1—figure supplement 1B and C*) (*Zhang et al., 2000*).

The rate of GEF-dependent Cdc42 activation was quantified by monitoring the displacement of GDP from Cdc42 in the presence of mant-GTP and unlabelled GMP-PNP, a non-hydrolysable GTP analogue. GDP to GTP exchange was measured both by the fluorescence intensity change of mant-GTP upon Cdc42 binding and by FRET between Cdc42 and mant-GTP. While both approaches gave comparable results, the FRET approach provided a higher signal-to-noise ratio. While neither Cdc42 alone nor Cdc42 and Bem1 displayed significant exchange of GDP for mant-GTP, the addition of Cdc24 provided robust GEF activity to Cdc42 (*Figure 1A*, blue line). The addition of Bem1 to the GEF reaction accelerated the rate of Cdc24 GEF activity further (*Figure 1A*, red line). The effect of Bem1-accelerated GEF activity was concentration dependent. Bem1 increased the observed rate constant of Cdc24 by 1.8x when present at 3 µM, and by 2.6x when present at 5 µM (*Figure 1B*). The C-terminal PB1 domain of Bem1, which interacts with the C-terminal PB1 domain of Cdc24 (*Ito et al., 2001*), was required for the stimulation of Cdc24 GEF activity, since a Bem1 mutant lacking this domain failed to augment GEF activity (*Figure 1—figure supplement 1D and E*). However, the PB1 domain of Bem1 alone was not sufficient to increase GEF activity (*Figure 1—figure supplement 1D and E*). These results demonstrate that Bem1 can directly increase Cdc24 GEF activity towards Cdc42 by interacting with the PB1 domain of Cdc24.

### Bem1 increases the rate and extent of Cdc24 phosphorylation by the PAK Cla4

Since Bem1 interacts with the PAK Cla4, which phosphorylates Cdc24 (*Gulli et al., 2000*; *Bose et al., 2001*), we next tested if Bem1 influenced the rate of Cdc24 phosphorylation by Cla4.

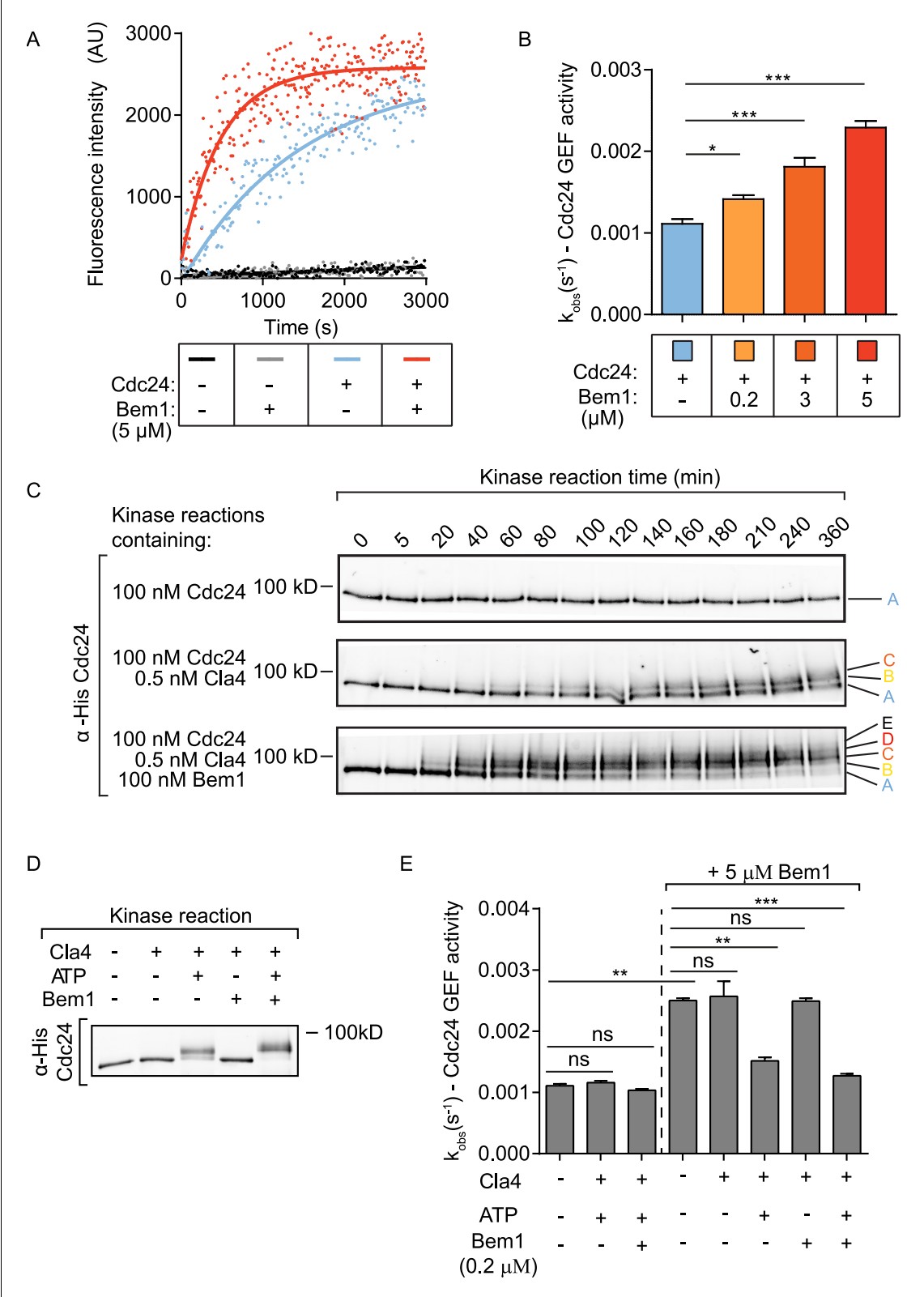

**Figure 1.** The scaffold Bem1 directly stimulates Cdc24 GEF activity in a reversible manner via PAK-dependent phosphorylation. (**A**) Fluorescence intensity change associated with the nucleotide exchange of GDP-Cdc42 for mant-GTP Cdc42. Fluorescence was measured after the addition of GDP-Cdc42 (9 µM) to reactions containing Mant-GTP (100 nM) and GMP-PNP (100 µM) in the absence (blue curve) and presence (red curve) of Bem1 (5 µM) and Cdc24 (60 nM). (**B**) Observed kinetic rate constants were obtained by fitting trace data to a single exponential equation. Error bars show SD and

*Figure 1 continued on next page*

*Figure 1 continued*

confidence where *p<0.05 and ***p<0.001. (C) In vitro kinase reactions in which the indicated proteins were incubated with Cdc24-6xHis. At the times indicated, samples were removed and analysed by SDS-PAGE and Western blotting using anti-His antibody to detect the electrophoretic mobility shift of Cdc24-6xHis. (D) A Western blot showing the phosphorylation of Cdc24 in the samples used for subsequent GEF assays. (E) The observed kinetic rate constants of Cdc24 GEF activity obtained by fitting trace data to a single exponential equation. The reactions indicated on the right had additional Bem1 added to 5 μM.

The following source data and figure supplements are available for figure 1:

**Source data 1.** Excel file showing the observed rate constants for the GEF assays presented in *Figure 1B and E*.

**Figure supplement 1.** The interaction of Cdc42 with mant- nucleotides and the dependence of scaffold stimulation on the PB1 domain.

**Figure supplement 1—source data 1.** Excel file showing the observed rate constants for the GEF assays presented in *Figure 1—figure supplement 1E*.

Kinase reactions were established in vitro to follow the kinetics of Cdc24 phosphorylation. While Cdc24 incubated with kinase assay buffer alone showed no phosphorylation-dependent electrophoretic mobility shift, addition of Cla4 resulted in a marked Cdc24 electrophoretic mobility shift in which three resolved forms of Cdc24 were apparent (*Figure 1C* top and middle panel). The inclusion of Bem1 in the kinase reaction accelerated the rate at which Cdc24 phosphorylation occurred, concomitant with the appearance of additional phosphorylated species of Cdc24 (*Figure 1C* bottom panel and *Figure 1—figure supplement 1F*). The electrophoretic mobility shift exhibited by Cdc24 in the presence of Cla4 and Bem1 in vitro was comparable to the electrophoretic mobility shift at the time of bud emergence in vivo, where Cdc24 is most abundantly phosphorylated (*McCusker et al., 2007*). These results indicate that Bem1 accelerates the rate and extent of Cdc24 phosphorylation by Cla4, generating hyperphosphorylated Cdc24.

## Phosphorylation of Cdc24 inhibits the scaffold-dependent increase in Cdc24 GEF activity in vitro

Our results identified three effects of Bem1 on Cdc24: Bem1 augments Cdc24 GEF activity, while also increasing the rate and extent of GEF phosphorylation by the PAK Cla4. Importantly, a recent study reported that phosphorylated Cdc24 displays reduced GEF activity (*Kuo et al., 2014*); however, this was interpreted to resolve the competition for polarity factors during polarity establishment, rather than being part of a Bem1-dependent regulatory mechanism. Scaffold-mediated augmentation of Cdc42 GTP production by the GEF may activate PAK activity to hyper-phosphorylate and subsequently inhibit the GEF, serving as a self-regulating mechanism. We therefore tested the effect of Cdc24 phosphorylation on scaffold-dependent Cdc24 GEF activity. Different samples of Cdc24 were generated: Cdc24 alone, Cdc24 phosphorylated by Cla4 and Cdc24 hyper-phosphorylated by Cla4 in the presence of Bem1. The Cdc24 samples incubated with kinase displayed a quantitative shift in electrophoretic mobility compared to the control, non-phosphorylated sample, indicating that the majority of Cdc24 is phosphorylated in these samples. In addition, the Cdc24 sample incubated with Cla4 and Bem1 migrated more slowly than that incubated with Cla4 alone and the electrophoretic mobility shifts in Cdc24 were dependent upon the addition of ATP (*Figure 1D*). The samples of Cdc24 were next assayed for GEF activity. Cdc24 that had been phosphorylated by Cla4 showed similar GEF activity to Cdc24 alone. However, Bem1 no longer stimulated GEF activity when Cdc24 was phosphorylated (*Figure 1E*). Importantly, the ability of Cla4 to antagonise Bem1 stimulation of Cdc24 GEF activity was dependent upon the presence of ATP in the reaction, indicating that Cla4's effect on GEF activity is mediated by phosphorylation rather than through a protein-protein interaction. Consistent with our previous observations, addition of Bem1 to 5 μM augmented the GEF activity of the non-phosphorylated Cdc24 sample; however, the GEF activity of the phosphorylated or hyper-phosphorylated Cdc24 were resistant to further Bem1 stimulation (*Figure 1E*).

## Phospho-regulation of Cdc24 impacts polarised growth in vivo

Our in vitro reconstitution experiments revealed that Bem1 stimulated Cdc24 GEF activity, while also attenuating this activation by inducing Cdc24 phosphorylation via the PAK Cla4. These results predict that non-phosphorylated Cdc24 would be active; as it would be constantly amenable to Bem1-stimulation, while constitutively phosphorylated Cdc24 would be less active, since it would be resistant to Bem1 stimulation. To begin to address these predictions, we first mapped phosphory-lated residues in Cdc24, before mutating these sites in vivo. Samples of Cdc24 were quantitatively phosphorylated in vitro by Cla4 alone or by Cla4 in the presence of Bem1, and then analysed by mass spectrometry. Cla4 phosphorylated 39 sites on Cdc24, of which 15 were only detected in the presence of Bem1. Since Cdc24 is also a target of Cdk1 (*Moffat and Andrews, 2004*; *McCusker et al., 2007*), we also mapped these sites after incubating Cdc24 with Cdk1-Cln2. This identified an additional seven sites. Thus, in total, we mapped 46 phosphorylation sites on Cdc24, while a previous study mapped 35 sites in vivo, 23 of which overlapped with those that we identified (*Figure 2—figure supplement 1A*) (*Wai et al., 2009*).

The mapped phosphorylation sites were mutated to alanine or aspartate and the constructs, together with a wild-type control, were integrated at the endogenous *CDC24* locus in vivo. Homologous recombination generated a phospho-mutant in which all 46 sites were mutated to alanine (*cdc24-46A*) and a phospho-mimetic mutant in which 28 sites were mutated to aspartate (*cdc24-28D*) (*Figure 2—figure supplement 1B*). We also generated mutants in which only the 15 Bem1-dependent phosphorylation sites in *CDC24* were mutated to alanine or aspartate. We refer to these mutants as *cdc24-15A* and *cdc24-15D*. We analysed Cdc24 protein mobility by Western blotting in the mutants and control cells during the cell cycle. Wild-type Cdc24 is phosphorylated at time zero due to the cell synchronisation with mating pheromone (*Gulli et al., 2000*). A portion of Cdc24 is then dephosphorylated until 15–30 min, at which point it undergoes additional phosphorylation at around 60 min after release from the G1-arrest, around the time at which cells form a bud (*Figure 2A*). Cell-cycle-dependent phosphorylation of Cdc24 was greatly diminished in the *cdc24-46A* mutant, indicating that many of the phospho-sites mapped by mass spectrometry in vitro are relevant in vivo. Conversely, the *cdc24-28D* mutant displayed constitutively reduced mobility by SDS-PAGE during the cell cycle that was similar to that of hyperphosphorylated wild-type Cdc24. The *cdc24-15A* and −15D mutants showed an intermediate level of phosphorylation between wild type and the *cdc24-46A/ cdc24-28D* mutants. In keeping with our in vitro biochemical observations, the *cdc24-46A* and the *cdc24-15A* mutants appeared to be active, since cells did not display an obvi-ous defect in the rate of colony formation at 37°C compared to control cells, while the *cdc24-28D* and the *cdc24-15D* mutants exhibited a strong defect in colony formation at 37°C (*Figure 2B*). The growth defect of *cdc24-28D* cells at 37°C was not due to degradation of the cdc24-28D protein, consistent with the idea that the mutations had not induced non-specific cdc24 mis-folding (*Figure 2—figure supplement 1C*). The temperature-dependent growth defect of the *cdc24-28D* mutant cells at 37°C was partially suppressed by the addition of sorbitol to the growth media. Sorbi-tol acts as an hyperosmotic stabilising agent and has previously been used to support the viability of *cdc24* mutants, which are defective in the polarised secretion of cell wall material (*Figure 2—figure supplement 1D*) (*Bender and Pringle, 1989*).

We also characterised polarised growth during the cell cycle, which is dependent upon cell cycle cues emanating from Cdk1 and Cdc42 activity (*Moseley and Nurse, 2009*; *McCusker and Kellogg, 2012*). After synchronous release into the cell cycle, samples were periodically removed and scored for the appearance of small buds, an indicator of polarised growth. By 60 min after release from the arrest, ~60% of wild-type cells had formed buds, while ~35% *cdc24-46A* mutants and only ~25% *cdc24-28D* mutants had formed buds, consistent with a defect in polarised growth due to defective regulation of Cdc24 phosphorylation (*Figure 2—figure supplement 1E* left panel). While it is pres-ently unknown why the *cdc24-46A* mutant displays a delay in bud emergence, the reduced rate of bud emergence in the *cdc24-28D* mutant did not appear to stem from the mutation of Cdk1-dependent sites in *CDC24*, since the *cdc24-15D* mutant in which only Bem1-dependent Cla4 sites were mutated also displayed a delay in bud emergence (*Figure 2—figure supplement 1E* right panel). Inspection of the morphology of the *cdc24-28D* mutant cells at 25°C revealed a significant propor-tion of cells that were large and unbudded, or displayed mis-shapen buds, indicative of defects in polarised growth due to aberrant Cdc24 phospho-regulation. The defects in morphology of the

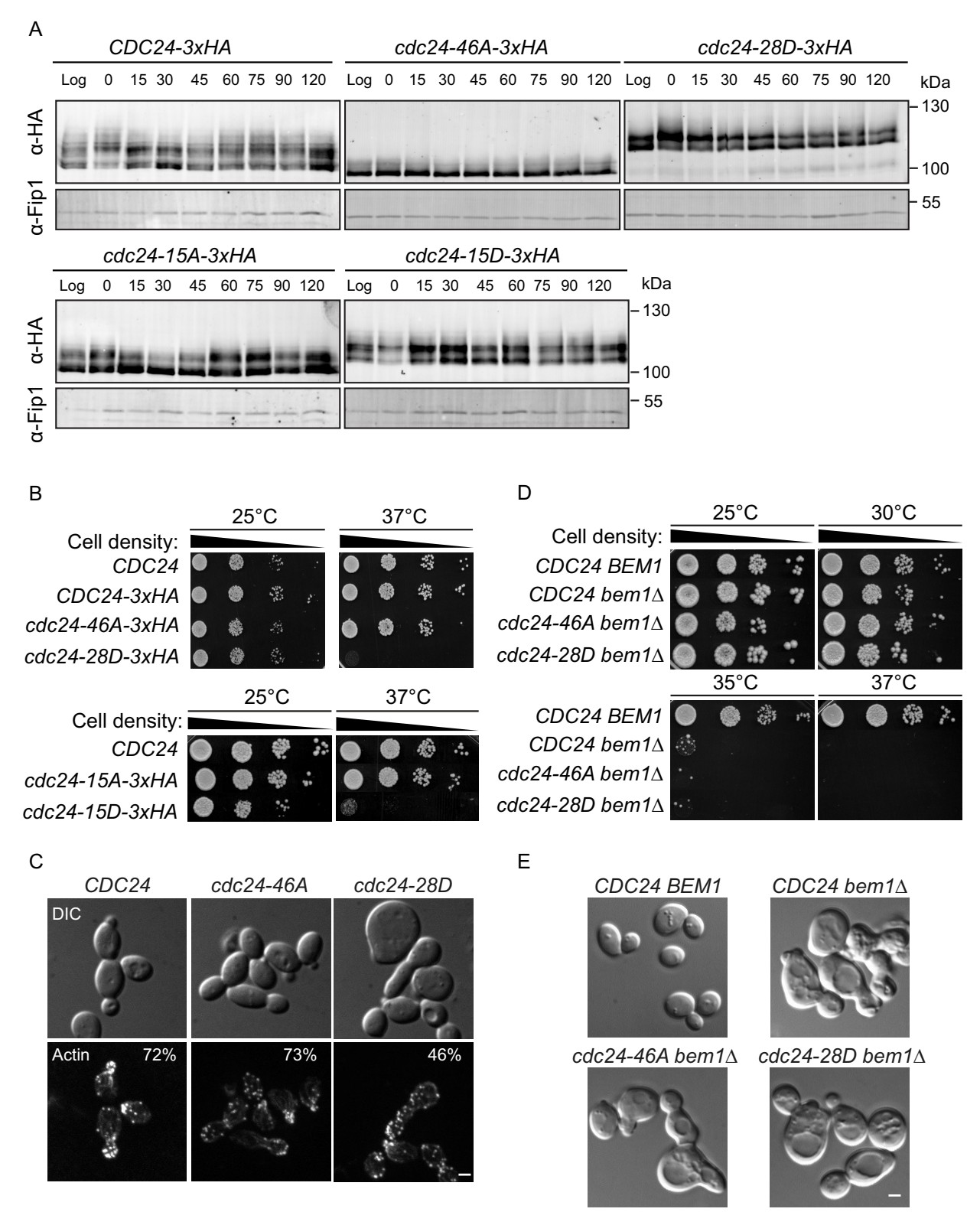

**Figure 2.** Phospho-regulation of Cdc24 is required for normal cellular polarity. (**A**) Phosphorylation of Cdc24-3xHA, cdc24-46A-3xHA, cdc24-28D-3xHA, cdc24-15A-3xHA and cdc24-15D-3xHA during the cell cycle. Cells were synchronised in mating pheromone and samples were removed at the times shown (top) and analysed by SDS-PAGE and Western blotting using anti-HA antibody to detect Cdc24 and anti-Fip1 antibody as a loading control. (**B**) cdc24-28D and cdc24-15D cells display a temperature-sensitive growth defect at 37°C. Serial dilutions of the cells indicated were spotted onto YPD

*Figure 2 continued on next page*

*Figure 2 continued*

plates and grown at 25°C and 37°C for 2 days. (C) The cells indicated were grown to mid-logarithmic phase at 25°C then fixed and stained with Alexa 546-phalloidin to visualise F-actin. DIC images are shown in the top and F-actin on the bottom panels. The numbers indicate the percentage of cells displaying polarised F-actin where more than 150 cells were counted for each strain. Scale bar: 2 μm. (D) The phenotype of the *cdc24* phosphorylation mutants is dependent on *BEM1*. Serial dilutions of the cells indicated were spotted onto YPD plates and grown at the temperatures indicated for 2 days. (E) DIC images showing the effect of deletion of *BEM1* in the *cdc24* phosphorylation mutants. Scale bar: 2 μm.

The following source data and figure supplements are available for figure 2:

**Figure supplement 1.** Mapped phosphosites on Cdc24 and mutant phenotypes.

**Figure supplement 1—source data 1.** Excel file showing the percentage cells of the indicated genotype displaying buds.

*cdc24-28D* mutant cells were reflected in their cell cycle doubling times (*CDC24* 99 min, *cdc24-46A* 111 min and *cdc24-28D* 140 min at 25°C in YPD medium). The morphological defects were also evident after F-actin visualisation (*Figure 2C*). While the actin cytoskeleton was polarised in ~75% of control and *cdc24-46A* cells, this was reduced to 46% in *cdc24-28D* cells. Moreover, the effect of the two *cdc24* phospho-mutant alleles was Bem1-dependent in vivo, since deletion of *BEM1* resulted in indistinguishable phenotypes in *CDC24*, *cdc24-46A* and *cdc24-28D* strains (*Figure 2D and E*).

We next examined the localisation of Cdc24 phospho-mutant proteins. As previously reported, wild-type Cdc24 localises to the nucleus in early G1 of the cell cycle, whereupon cell cycle cues trigger its export to the cytoplasm and localisation to the site of polarity establishment (*Figure 3A*) (*Toenjes et al., 1999*; *Nern and Arkowitz, 2000*; *Shimada et al., 2000*). Importantly, the cdc24-46A protein was enriched at the site of polarity establishment compared to the wild-type protein (*Figure 3A and B*), while the cdc24-28D, although polarised, displayed less enrichment at the pole and prominent nuclear localisation at points in the cell cycle when the wild-type protein had been exported from the nucleus to the cytoplasm (*Figure 3A and B*). Cdc24 is sequestered in the nucleus during early G1 of the mitotic cell cycle via its interaction with Far1 (*Nern and Arkowitz, 2000*; *Shimada et al., 2000*), a Cdk1 inhibitor that is degraded late in G1 by rising cyclin-Cdk1 activity and the proteasome, thus releasing Cdc24 into the cytoplasm (*Peter et al., 1993*; *Henchoz et al., 1997*). We have not yet successfully deleted *FAR1* in the *cdc24* phospho-mutant background to test the effect on the phenotype of the *cdc24* phospho-mutants. The expression of the mutants and wild-type protein were comparable (*Figure 3—figure supplement 1A*). The mutation of the phosphorylation sites did not appear to result in complete Cdc24 delocalisation, but rather altered its dynamics at sites to which the wild-type protein normally localises. Consistently, Cdc42 was also enriched at the cell pole in the *cdc24-46A* mutant (*Figure 3C*). As a more direct test of whether Cdc42 GTP levels were altered in the *cdc24* phosphorylation mutants, we imaged gic2$_{1-208}$-yEGFP, which includes a CRIB motif that has previously been used as a marker for Cdc42 GTP (*Tong et al., 2007*). Consistent with the localisation of Cdc24 and Cdc42, we found higher levels of the Cdc42 GTP reporter in the *cdc24-46A* and the *cdc24-15A* mutants compared to control cells, while the reporter was less enriched at the pole in *cdc24-28D* and *cdc24-15D* mutants (*Figure 3D*).

These results indicate that the phospho-blocking and phospho-mimetic mutations confer distinct phenotypes and are thus likely to affect Cdc24 in a specific manner. As a means of corroboration, we tested the GEF activity of purified cdc24-46A and −28D proteins in the presence and absence of Bem1. While both mutants exhibited similar basal GEF activity, Bem1 was able to stimulate the cdc24-46A mutant, but not that of the cdc24-28D mutant (*Figure 3E* and *Figure 3—figure supplement 1B*). The lower effect of Bem1 on cdc24-28D did not appear to stem from markedly reduced affinity of cdc24-28D for Bem1, since similar levels of Bem1 associated with Cdc, cdc24-46A and cdc24-28D, as quantified from pull-down experiments (*Figure 3—figure supplement 1C and D*). In summary, our in vivo data indicate that the *cdc24-46A* mutant that displays reduced phosphorylation is viable and is enriched at the cell pole, together with its cognate GTPase Cdc42. Conversely, the *cdc24-28D* mutant, which displays constitutively reduced electrophoretic mobility, results in a temperature-sensitive phenotype in vivo, reduced recruitment of the mutant protein to the cell pole and aberrant cell polarity. We reasoned that the increased recruitment of cdc24, Cdc42 and the Cdc42 GTP reporter to the cell pole in the *cdc24-46A* mutant might reflect increased Cdc42 GTP

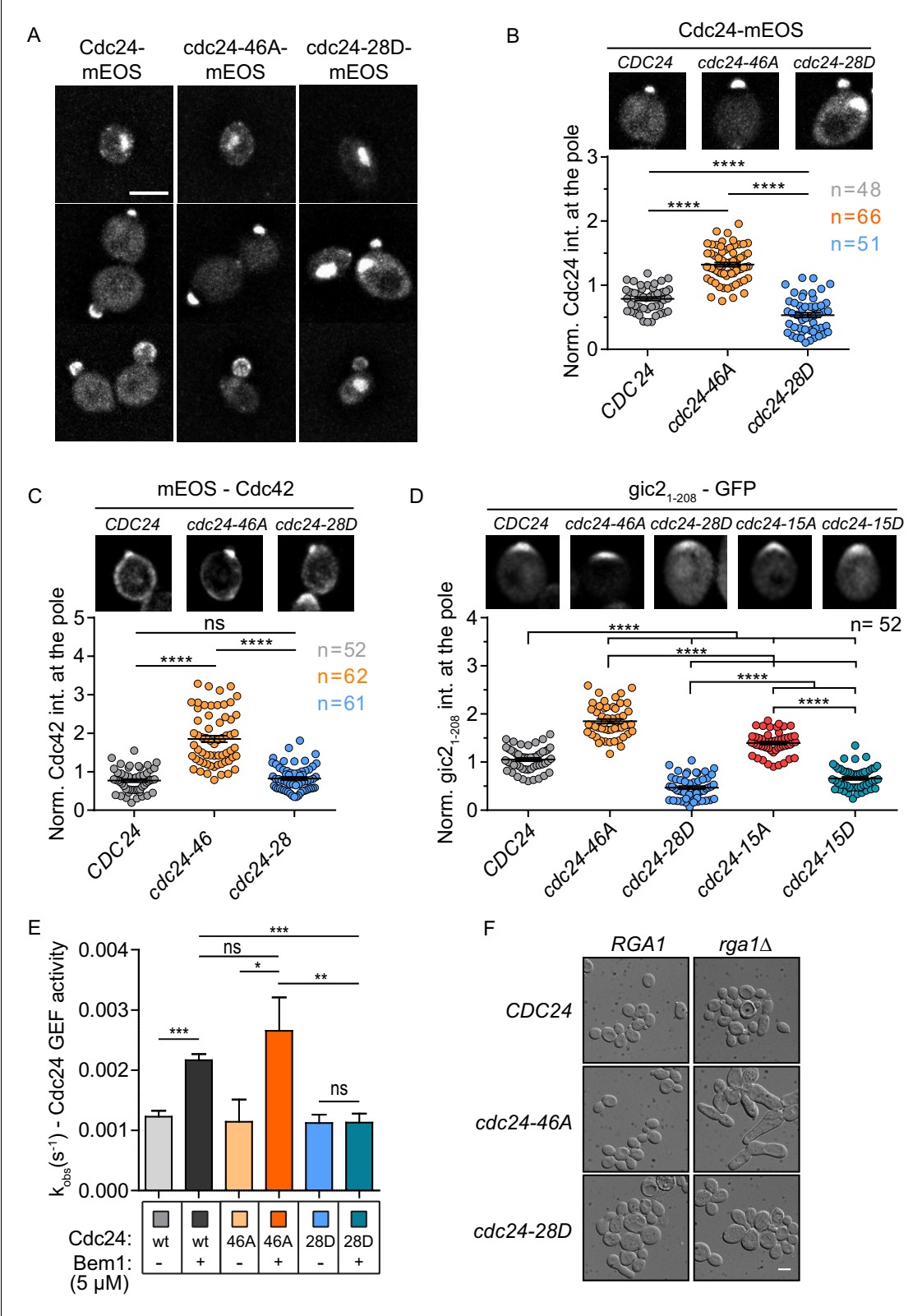

**Figure 3.** Phospho-regulation of Cdc24 is required for normal GEF localisation at the cell pole in vivo and Bem1-stimulated GEF activity in vitro. (**A**) Maximum projection images of deconvolved z-stacks. Scale bar: 5 μm. (**B**) The average fluorescence intensity signal of Cdc24 at the cell pole was plotted for 48–66 cells of the indicated strains. Error bars show SD and confidence where ****p≤0.0001. (**C**) Maximum projection images of deconvolved z-stacks showing mEOS-Cdc42 fluorescence and quantification of the average fluorescence intensity. The significance of error bars is the

*Figure 3 continued on next page*

*Figure 3 continued*

same as in (B). (D) Maximum projection images of deconvolved z-stacks showing gic2$_{1-208}$-yEGFP fluorescence and quantification of the average fluorescence intensity. The significance of error bars is the same as in (B) and (C). (E) Observed kinetic rate constants of Cdc24 GEF activity for cdc24-46A and cdc24-28D +/− Bem1. Note how Bem1 stimulates the rate of cdc24-46A GEF activity, but not that of cdc24-28D. Error bars show SD and confidence where *p<0.05 and **p≤0.01. (F) DIC images showing the effects of *RGA1* deletion on the indicated strains. Scale bar: 5 μm.

The following source data and figure supplements are available for figure 3:

**Source data 1.** Excel file showing the source data for *Figure 3*, including the normalised Cdc24-mEos intensity, the normalised mEos-Cdc42 intensity, the normalised gic2 (1-208)-yEGFP intensity and the observed rate constants for Cdc24 GEF activity.
**Figure supplement 1.** Expression of Cdc24-mEOS in vivo and the activity of cdc24 phospho-mutants in vitro.
**Figure supplement 1—source data 1.** Excel file showing the band intensity of the data presented in *Figure 3—figure supplement 1D*.

production, but that the mutant might not exhibit morphological defects because Cdc42 GAPs may buffer the GTPase module. We therefore deleted the GAP *RGA1* in control and cdc24 phospho-mutants to test whether loss of GAP buffering would result in morphological defects. Indeed, *cdc24-46A* mutants were sensitive to a reduction in GAP activity compared to *CDC24* and *cdcd24-28D* cells. The *cdc24-46A rga1△* double mutants were highly elongated, consistent with excessive Cdc42 GTP production, while the *cdc24-28D rga1△* mutant cells were not elongated (*Figure 3F*). Consistently, the deletion of Cdc42 GAPs and *CLA4* also results in highly elongated cells (*Caviston et al., 2003*). These in vivo results support our biochemical model derived from in vitro reconstitution experiments in which phosphorylation of Cdc24 attenuates the scaffold-mediated augmentation of GEF activity.

## Discussion

Scaffold proteins contribute to dynamic aspects of signalling, including signal amplitude, feedback and localisation. Here, we show that the scaffold Bem1 in budding yeast plays a direct role in stimulating GEF activity during Cdc42 activation. This spike in Cdc42 GTP production is likely to contribute to the increase in Cdc42 activation that ensues during polarity establishment in vivo (*Figure 4*) (*Gulli et al., 2000*; *Wai et al., 2009*; *Atkins et al., 2013*; *Smith et al., 2013*). The augmentation of GEF activity occurs in a reversible fashion, since the scaffold increases the rate of GEF phosphorylation by the PAK Cla4, which attenuates scaffold-mediated GEF activation. Thus, phosphorylation may provide the GEF with something akin to a molecular memory of previous scaffold encounters, generating a pulse of GEF activity that contributes to flux through the Cdc42 GTPase module. It will be important to identify the phosphatase that antagonises Cdc24 phosphorylation by the PAK, Cla4, and to understand the mechanisms that determine the kinase-phosphatase activity during polarity establishment.

Previous studies in *U. maydis* and *S. pombe* reported that phosphorylation regulates Cdc42 GEFs by promoting its degradation, or by its 14-3-3-dependent sequestration, respectively (*Frieser et al., 2011*; *Das et al., 2015*). While a previous study found that the PAK Cla4 played a positive role in polarity establishment in *S. cerevisiae* (*Kozubowski et al., 2008*), subsequent work found that phosphorylation of Cdc24 by Cla4 resulted in reduced GEF activity, while increased Cdc42 recruitment to the cell pole was observed in Cdc24 phosphorylation mutants (*Kuo et al., 2014*; *Wu et al., 2015*). Our study supports these latter observations, additionally linking phosphorylation to a scaffold-dependent mechanism that promotes the localised activation of Cdc42 via Cdc24 regulation, an enigmatic problem in the field (*Smith et al., 2013*; *Woods et al., 2015*). We found that phosphorylation of Cdc24 appeared to reset GEF activity to a basal state, rather than inhibiting it altogether, potentially allowing the system to remain responsive to activation from other signalling sources. Previous studies had not reported strong morphological defects when Cdc24 phosphorylation sites were mutated (*Wai et al., 2009*; *Kuo et al., 2014*). This may stem from our observation that Cdc42 GAPs play an important role in counteracting the increased Cdc42 GTP levels that accumulate when Bem1 stimulates unphosphorylated Cdc24. Presently, it is unknown if Bem1 augments Cdc24 GEF

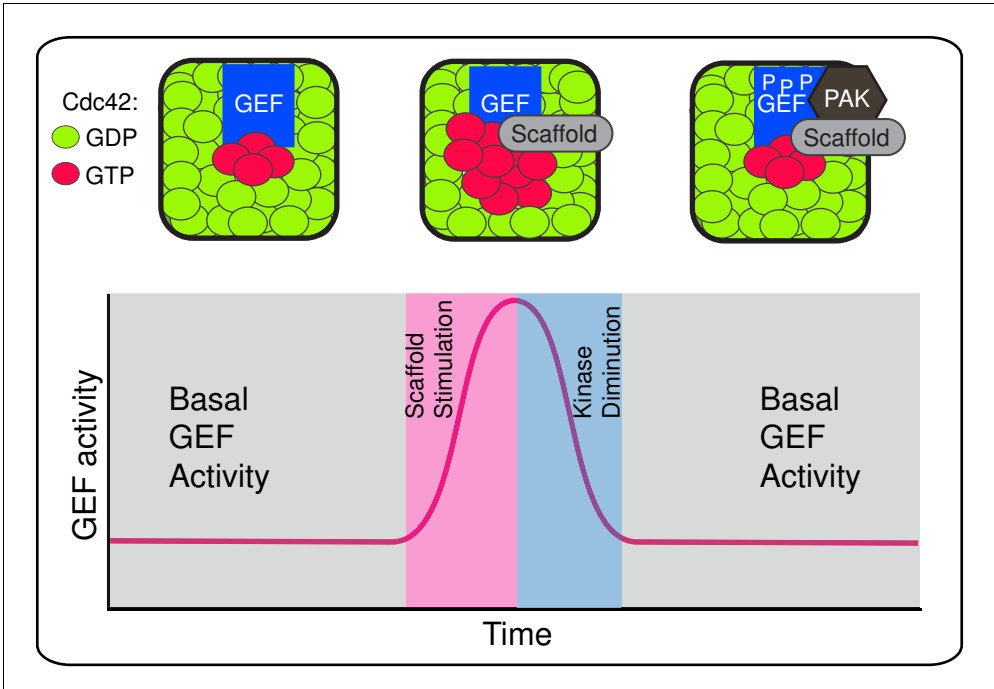

**Figure 4.** A working model depicting scaffold-mediated gating of Cdc42 signalling flux.

activity by tethering constituents of the GTPase module, hence increasing the local concentration of Cdc42 GDP and Cdc24, or by acting allosterically on the GEF; however, the report that Bem1 interacts preferentially with Cdc42 GTP rather than Cdc42 GDP would be consistent with the latter model (*Bose et al., 2001*). It will also be important to test whether any scaffold that increases the local Cdc24-Cla4 concentration would stimulate Cdc24 phosphorylation, or whether Bem1 exerts specific effects.

Our observation that the scaffold directly stimulates GEF activity is consistent with two recent reports. First, that scaffold-deleted cells follow an evolutionary pathway in which selective pressure down-regulates Cdc42 GAP activity (*Laan et al., 2015*). We propose that the evolutionary-driven down-regulation of GAP activity is likely to counteract the diminution in Cdc42 GTP levels that would ensue from loss of the scaffold. Second, it was reported that scaffold-deleted cells display reduced activity of a Cdc42 GTP biosensor in vivo (*Smith et al., 2013*). Our results are consistent with these observations and indicate that Bem1 is able to directly stimulate Cdc24 GEF activity towards Cdc42.

## Materials and methods

### Plasmid construction

*BEM1*, *bem1ΔPB1* (residues 1 to 466) and *bem1-PB1* (residues 469 to 550) were amplified by PCR with NdeI and XhoI restriction sites and cloned into a modified pGEX6P-2 vector, in which the BamHI site in the multiple cloning site was changed to NdeI. This generated pDM256, pDM516 and pDM502, respectively, in which the Bem1 constructs were tagged at the N-terminus with GST that could be subsequently removed by digestion with rhinovirus 3C protease.

*CDC24* was amplified by PCR with NdeI and XhoI sites and ligated into pET21a to generate pDM272, in which Cdc24 is C-terminally tagged with 6xHis. The nucleotide sequence encoding amino acids 75–854 were synthesised with HindIII and KpnI restriction sites (Bio-Basic, Markham, Canada). Four constructs were synthesised: one in which the 46 serines or threonines mapped by mass spectrometry were mutated to alanine (pDM615), one in which the 46 phosphosites were mutated to aspartic acid (pDM616), one in which only the 15 Bem1-specific phosphosites were mutated to alanine (pDM780) and one in which the 15 Bem1-specific sites were mutated to aspartic

acid (pDM781). The HindIII-KpnI fragments were cloned into pDK51, a *URA3* yeast vector enabling the in-frame fusion of the *cdc24* mutant fragments with three copies of the HA epitope to generate pDM630, pDM632, pDM782 and pDM784 (*Carroll et al., 1998*). Oligonucleotides with homology to *CDC24* were then used to generate *cdc24-46A-3xHA::URA3*, *cdc24-46D-3xHA::URA3*, *cdc24-15A-3xHA::URA3* and *cdc24-15D-3xHA::URA3* PCR products that were transformed into yeast. The *cdc24* mutants were sequenced in the resulting yeast transformants, indicating that homologous recombination at the *CDC24* locus had generated a *cdc24-46A-3xHA* mutant (DMY2151), a *cdc24-28D-3xHA* mutant (DMY2154), a *cdc24-15A-3xHA* mutant (DMY2333) and a *cdc24-15D-3xHA* mutant (DMY2134). Since the *cdc24-46D* mutant was not recovered, this mutant may not be viable. The *cdc24-46A* and *cdc24-28D* mutants were amplified by PCR from genomic DNA of DMY2151 and DMY2154 yeast strains, respectively, introducing NotI and XbaI sites, in addition to a ribosome-binding site. These PCR products were cloned into pET21a, generating pDM645 and pDM647, respectively, in which the cdc24 phospho-mutants were C-terminally tagged with 6xHis.

To monitor the in vivo localisation of the Cdc24 phospho-mutants, *CDC24*, *cdc24-46A* and *cdc24-28D* were amplified by PCR from genomic DNA prepared from DMY2147, DMY2151 and DMY2154 yeast, introducing HindIII and SpeI sites. The PCR products were cloned into a modified pRS416 plasmid, consisting of a *CYC1* promoter between KpnI and SalI sites. Subsequently, mEOS, in addition to an amino-terminal GAGAGG linker and a carboxyl-terminal GAGAG linker-6xHis-tag, were added to the C-terminus of the constructs using SpeI and NotI sites. Finally, an *ADH1* terminator sequence was introduced using a NotI site, generating pDM700, pDM701 and pDM704 for *CDC24*, *cdc24-46A* and *cdc24-28D*, respectively.

To monitor the in vivo level *and* localisation of Cdc42 GTP in the control and *cdc24* phospho-mutant strains, we generated a gic2$_{1-208}$-yEGFP fusion, containing a CRIB motif that was cloned into a modified pRS413 plasmid consisting of a *CYC1* promoter (pDM842) (*Curran et al., 2014*). Full-length wild-type *CDC42* was amplified with oligonucleotides that introduced EcoRI and AgeI sites, in addition to an N-terminal 6xHis-6xGly linker and a Strep-tag II sequence. The PCR product was cloned into a modified pPICZa vector (Thermo Fisher Scientific, Waltham, USA) to generate pDM401 for expression of the fusion from the *AOX1* promoter in *Pichia pastoris*. The pRS315 *CDC42*prom-mEOS-GAGA-*CDC42* plasmid (pDM303) was constructed in three steps: the *CDC42* promoter was cloned into pRS315 on a SalI-BamHI fragment. The *CDC42* ORF and terminator were then ligated into this vector using an NdeI site that was engineered 5' of the BamHI site and a 3' SacII site. Finally, mEOS was ligated into the NdeI digested plasmid and checked for the correct orientation. All plasmids were verified by sequencing.

## Yeast growth conditions

Yeast strains were generated and grown using standard genetic techniques. Cells were grown in selective or rich media, depending upon the experiment. For cell viability assays, yeast were grown in rich medium at 25°C until mid-log phase. Ten-fold serial dilutions were spotted onto YPD plates, which were incubated at the indicated temperatures for 2 days.

## Protein expression and purification

Nucleotide-free and GDP-loaded 6xHis-Strep-tag-II-Cdc42 was purified in a two-step affinity purification method by Co$^{2+}$-immobilised metal affinity chromatography (IMAC), followed by Strep-Tactin affinity chromatography. Ground yeast powder was generated using a Retsch PM100 grinding mill (Retsch, Haan, Germany) and a 125 ml steel bowl with 20 mm ball bearings chilled with liquid nitrogen. Greater than 85% lysis was typically observed after eight disruption cycles. For Cdc42 purification, room temperature lysis buffer (20 mM Tris-HCL (pH = 8.0), 1 M NaCl, 5% glycerol, 0.5% CHAPS supplemented with EDTA-free Protease inhibitor cocktail (Roche, Basel, Switzerland) and 1 mM PMSF) was added when the yeast powder showed signs of melting. The lysate was then stirred immediately for 10 min at 4°C and remained at 4°C for the subsequent purification. The cell lysate was centrifuged at 48,000 x *g* for 30 min in a JA25.5 rotor (Beckman, Brea, USA), and the supernatant was loaded into a Co$^{2+}$-IMAC column. After extensive washing (20 mM Tris-HCL (pH = 8.0), 1 M NaCl, 5% glycerol and 0.05% Tween-20), 6xHis-Strep-tag-II-Cdc42 was eluted in Co$^{2+}$-IMAC elution buffer (20 mM Tris-HCL (pH = 8.0), 150 mM NaCl, 5% glycerol and 250 mM imidazole and 0.05% Tween-20). In the second purification step, Strep-Tactin Superflow Plus (Qiagen, Venlo, The

Netherlands) was added to the elution and agitated for 30 min at 4°C. The sample was loaded onto a column and then washed extensively with Strep-Tactin wash buffer (20 mM Tris-HCL (pH = 8.0), 150 mM NaCl, 5% glycerol and 0.05% Tween-20). Nucleotide-free Cdc42 was generated by washing the column with 50 column volumes of Strep-Tactin wash buffer supplemented with 25 mM EDTA, followed by an additional washing step using Strep-Tactin wash buffer. GDP-bound Cdc42 was generated by washing the column with Strep-Tactin wash buffer supplemented with 200 µM GDP and 10 mM $MgCl_2$, then incubating for 1 hr at room temperature. The non-bound GDP and $Mg^{2+}$ were removed in an additional washing step. GDP-loaded 6xHis-Strep-tag-II-Cdc42 was eluted in Strep-Tactin elution buffer (20 mM Tris-HCL (pH = 8.0), 150 mM NaCl, 5% glycerol supplemented with 5 mM d-Desthiobiotin and 0.05% Tween-20). Nucleotide-free Cdc42, which was produced for nucleotide binding experiments, was eluted in elution buffer before the GDP loading step. Protein concentration was estimated by absorption at 280 nm using $\varepsilon = 21430$ $M^{-1}cm^{-1}$ as the calculated molar excitation coefficient. Samples were aliquoted in 50 µl volume and flash frozen in liquid nitrogen for storage.

Active Cla4-TAP was purified from 4 L of budding yeast after lysis in liquid nitrogen using a rotary mill, as described for the Cdc42 purification. Chilled, lysed yeast powder was resuspended in three volumes of lysis buffer (10 mM Tris-HCL (pH = 8.0), 1 M NaCl, 0.1% IGEPAL, 50 mM NaF, 2 mM $Na_3VO_4$, 100 mM $\beta$-glycerol phosphate, 200 mM potassium acetate and 1 mM PMSF). After centrifugation at 110,000 x $g$ for 90 min, the supernatant was incubated with 300 µl IgG Sepharose for 2 hr. The beads were loaded onto a 1 ml column and washed three times in 10 ml lysis buffer then in 10 ml TEV buffer (10 mM Tris-HCL (pH = 8.0), 150 mM NaCl, 0.1% IGEPAL, 0.5 mM EDTA and 1 mM DTT). TEV protease was then added overnight at 4°C. The eluate was diluted three-fold in calmodulin binding buffer (10 mM Tris-HCL (pH = 8.0), 150 mM NaCl, 0.1% IGEPAL, 1 mM magnesium acetate, 1 mM imidazole, 2 mM $CaCl_2$, and 10 mM $\beta$-mercaptoethanol), then incubated with 300 µl calmodulin Sepharose for 2 hr. The column was washed three times in 10 ml calmodulin binding buffer then eluted in kinase assay buffer (50 mM Hepes (pH = 7.6), 2 mM $MgCl_2$, 0.05% Tween-20, 1 mM DTT, 10% glycerol and 2 mM EGTA). The eluted kinase was dialysed in kinase assay buffer lacking EGTA then flash frozen in liquid nitrogen for storage.

Cdc24-6xHis and GST-tagged Bem1 were expressed in Bl21-CodonPlus (DE3) cells. Briefly, cells were grown in terrific broth at 37°C until an $OD_{600}$ ~2 (*Tartoff and Hobbs, 1987*). Expression was induced by the addition of IPTG to 0.2 mM, after which cells were grown for 12 hr at 18°C. Cells were harvested and flash frozen in liquid nitrogen. For Cdc24-6xHis purification, room temperature lysis buffer (20 mM Tris-HCL (pH = 8.0), 1 M NaCl, 5 mM imidazole supplemented with EDTA-free Protease inhibitor cocktail (Roche, Basel, Switzerland) and 1 mM freshly prepared PMSF) was added to chilled, ground bacterial powder. The lysate was immediately stirred at 4°C for 10 min and sonicated for 30 s three times. The cell lysate was centrifuged at 48,000x $g$ for 30 min, and the supernatant was loaded into a $Co^{2+}$-IMAC column. After extensive washing (20 mM Tris-HCL (pH = 8.0), 1 M NaCl, 5 mM imidazole), Cdc24-6xHis was eluted (20 mM Tris-HCL (pH = 8.0), 150 mM NaCl, 500 mM imidazole). The protein was dialysed twice (20 mM Tris-HCL (pH = 8.0), 150 mM NaCl) and flash frozen in liquid nitrogen for storage.

For the Bem1 purification, cells were lysed by adding room temperature lysis buffer (50 mM Tris-HCL (pH = 7.5), 1 M NaCl, 0.1% Tween-20 and 5 mM DTT, an EDTA-free Protease inhibitor cocktail tablet and 1 mM freshly prepared PMSF) to ground bacterial powder. The lysate was processed by sonication and centrifugation in the same manner as the Cdc24 −6xHis lysate. The supernatant was loaded onto a glutathione column. The column was extensively washed (50 mM Tris-HCL (pH = 7.5), 250 mM KCl, 0.1% Tween-20 and 5 mM DTT), and then GST-Bem1 was eluted (50 mM Tris-HCL (pH = 7.5), 250 mM KCl and 5 mM reduced glutathione). For GST pull-down experiments, GST-Bem1 was dialysed twice against 20 mM Tris-HCL (pH = 7.5), supplemented with 150 mM NaCl and flash frozen in liquid nitrogen for storage. For enzymatic experiments, GST-Bem1 was dialysed extensively in protease buffer (20 mM Tris-HCL (pH = 7.5), 150 mM NaCl, 1 mM DTT and 1 mM EDTA), the GST was digested using rhinovirus 3C protease then removed by re-loading the protein onto a glutathione agarose column. The flow-through, which contained Bem1, was dialysed extensively in 20 mM Tris-HCL (pH = 7.5), 150 mM NaCl then flash frozen in liquid nitrogen for storage.

## Cdc24 GEF assay

Förster resonance energy transfer (FRET) between Cdc42 and N-methylanthraniloyl-GTP (mant-GTP) was measured to monitor the Cdc24-mediated GDP to mant-GTP exchange reaction on Cdc42 in real time. Trp97 of Cdc42, which is in close proximity to the GTP binding site, was excited using 280 nm wavelength light using a 5 nm bandwidth. The FRET signal was detected at the emission peak of mant-GTP, at 440 nm using an 8 nm bandwidth. All fluorescence measurements were performed at 27°C on a Tecan Infinite M1000PRO plate reader (Tecan Group, Männedorf, Germany) in 384-well, non-binding microplates (Greiner Bio-One, Courtaboef, France), in a 10 µl reaction volume. The final buffer conditions were 20 mM Tris-HCl (pH = 8.0), 150 mM NaCl, 1 mM DTT, 5 mM MgCl$_2$, 100 nM mant-GTP, supplemented with 100 µM GMP-PNP nucleotide to maintain a low concentration of the mant-GTP fluorophore. Cdc24 was used at a final concentration of 60 nM after 30 min room temperature pre-incubation with Bem1 at 5 µM. The reaction was started by adding GDP-bound Cdc42 to 9 µM final concentration and exchange was monitored for at least 2000 s with 15 s intervals. For each sample, a mock reaction was used in the absence of GDP-loaded Cdc42 to normalise for bleaching and to subtract possible sources of background noise such as Cdc24-mant-GTP interaction. The intrinsic GDP to mant-GTP exchange rate of Cdc42 was determined in the absence of Cdc24.

A single exponential equation (*Equation 1*) was fitted to each kinetic trace using GraphPad Prism 6 version 6.05 and the observed kinetic rate constants were compared.

$$I = I_{Max} + (I_{Max} - I_{Min})(1 - e^{-k_{obs}t}),$$ (1)

where $I$ is the fluorescence intensity change, $I_{Max}$ is the maximal fluorescence intensity, $I_{Min}$ is the minimal fluorescence intensity, $k_{obs}$ is the observed kinetic rate constant and $t$ is the time in seconds.

All reactions were performed at least three times with proteins from two different purifications.

## Mant-nucleotide binding

A 100 nM final concentration of mant-GDP or mant-GTP were titrated with increasing concentration of nucleotide-free Cdc42 in buffer containing 20 mM Tris-HCl (pH = 7.5),150 mM NaCl and 1 mM DTT at 27°C. The interaction was monitored for 30 min by FRET at 440 nm using 280 nm wavelength light for excitation. Maximal fluorescence changes (-) were obtained by fitting a single exponential onto each trace:

$$I = I_{Max} + (I_{Max} - I_{Min})(1 - e^{-k_{obs}t}),$$

where $I$ is the fluorescence intensity change, $I_{Max}$ is the maximal fluorescence intensity, $I_{Min}$ is the minimal fluorescence intensity, $k_{obs}$ is the observed kinetic rate constant and $t$ is the time in seconds.

Maximal fluorescence changes of each trace were plotted against the nucleotide-free Cdc42 concentration and the dissociation constants were determined by fitting a quadratic binding equation:

$$I = \frac{(K_d + [Cdc42] + [mant]) - \sqrt{(K_d + [Cdc42] + [mant])^2 - 4[Cdc42][mant]}}{2[mant]},$$

where $I$ is the maximal fluorescence change observed in each trace, $K_d$ is the dissociation constant, $[Cdc42]$ is the total Cdc42 concentration and $[mant]$ is the total mant-GDP or total mant-GTP concentration.

All reactions were performed at least three times with proteins from two different purifications.

## In vitro kinase assays

For the time-resolved kinase reactions shown in *Figure 1C*, a 100 nM final concentration of Cdc24 was incubated with 500 pM Cla4 in the presence or in the absence of 100 nM Bem1 in kinase buffer (20 mM Tris (pH = 8.0), 2 mM MgCl$_2$, 10% glycerol and 1 mM ATP) at 30°C. SDS loading buffer was added to samples at the indicated times and analysed by SDS-PAGE and Western blotting using mouse anti-His antibody (Euromedex, Souffelweyersheim, France) and AlexaFluor532 goat anti-mouse IgG(H+L) (Life technology, Carlsbad, USA) as primary and secondary antibodies, respectively. Fluorescence was detected using a Typhoon TRIO+ scanner (GE Healthcare). The result was

analysed by ImageQuant TL 1D version 8.1 software (GE Healthcare). The time-resolved kinase reactions were repeated at least three times.

## In vitro kinase reaction for the GEF assay

A 600 nM final concentration of Cdc24 was treated with 3 nM Cla4 in the presence or absence of 2 µM Bem1 - under these conditions more than 80% of Cdc24 is estimated to exist in a complex with Bem1 - in kinase buffer (20 mM Tris (pH = 8.0), 2 mM $MgCl_2$, 10% glycerol and 1 mM ATP) at 30°C for 2 hr. Phosphorylation of Cdc24 was validated by Western blot using anti-His antibody. A 60 nM final concentration of phosphorylated samples were used directly in the GEF assay, as described above.

## Cell synchronisation, Western blotting and actin staining

For cell synchronisation, 0.5 µg/ml α-factor was added to mid-log phase yeast cultures at 25°C for 3 hr, as described previously (*McCusker et al., 2007*). Cells were then washed three times in YPD lacking pheromone for synchronised release into the cell cycle at 25°C. To determine the budding index, samples were removed at the times indicated, fixed by the addition of formaldehyde to 3.7%, washed with PBS then at least 200 cells were scored per time point to ascertain the percentage of budded cells.

To characterise the mobility of Cdc24 by SDS-PAGE during the cell cycle, 1.5 ml cells were removed at the indicated time points. After centrifugation, small glass beads were added to the pellet and the samples were flash frozen in liquid nitrogen. Samples were vigorously agitated with glass beads in 100 µl SDS loading buffer supplemented with 1 mM PMSF, 50 mM NaF and 75 mM $\beta$-glycerol phosphate. Samples were immediately boiled and analysed by SDS-PAGE and Western blotting. For actin staining, yeast were grown to early log phase, fixed and then stained with Alexa 546-phalloidin, as previously described (*Jose et al., 2015*).

## GST-pull down

Equal amounts of Cdc24, cdc24-46A and cdc24-28D were added to excess amounts of GST or GST-Bem1 immobilised on glutathione agarose beads (Sigma Aldrich, St. Louis, UA). After incubation on ice for 1 hr, beads were washed extensively (20 mM Tris pH = 7.5, 150 mM NaCl and 0.1% Tween-20) and proteins were eluted (20 mM Tris pH = 7.5, 150 mM NaCl, 0.1% Tween-20 and 5 mM reduced glutathione). Proteins were then TCA precipitated, and analysed by SDS-PAGE and Western blotting. All pull-down experiments were repeated three times.

## Phosphorylation site mapping

In order to identify phospho-sites on Cdc24, phospho-peptides of Cdc24 after trypsin digestion were enriched by strong cation exchange chromatography, followed by IMAC, and then analysed by mass spectroscopy. For kinase reactions, a 600 nM final concentration of Cdc24 was treated with 3 nM Cla4 in the presence or absence of 2 µM Bem1 in kinase assay buffer (20 mM Tris (pH = 8.0), 2 mM $MgCl_2$, 10% glycerol and 1 mM ATP) in a 25 µl reaction volume at 30°C for 45 min. Under these conditions in the presence of Bem1, maximal Cdc24 phosphorylation was observed. After the kinase reaction, disulphide bonds in the proteins were reduced by adding DTT to a final concentration of 5 mM at 56°C for 25 min. Samples were then cooled to room temperature before alkylating cysteines at room temperature for 30 min in the dark in 14 mM of freshly prepared iodoacetamide. Excess iodoacetamide was quenched by adding DTT to 5 mM and incubating at room temperature for 15 min. In order to isolate phosphorylated Cdc24, 0.5 µg protein was analysed by 12% SDS-PAGE and Coomassie-stained Cdc24 bands were excised for trypsin digestion.

## LC-MS/MS

Excised gel bands were sliced into approximately 1 mm cubes, and destained by incubation in 50% acetonitrile/50 mM ammonium bicarbonate at 37°C. Destained gel slabs were dehydrated by incubation in neat acetonitrile, and re-hydrated in digestion buffer (50 mM ammonium bicarbonate, 10 µg/ml trypsin). Digestions were incubated overnight at 37°C. The digestion supernatant was decanted to a fresh tube, and the gel pieces were washed two times with 50% acetonitrile/50 mM ammonium bicarbonate. The washes were combined with the digestion supernatant, and evaporated via

SpeedVac (Thermo Fisher Scientific, Waltham, MA). Digests were resuspended in 5% formic acid/5% acetonitrile, and cleaned via c18 microextraction (*Rappsilber et al., 2003*). Samples were loaded onto a 40-cm fused silica column (100 µM ID) packed in-house with c18 resin (1.8 µm particle size) and analysed on an Orbitrap Fusion Tribrid mass spectrometer operating in data dependent mode, using a 90 min gradient over which acetonitrile in 0.1% formic acid was increased from 3% to 40%. Proteins present within each sample were identified by SEQUEST search against the yeast proteome (Uniprot). Peptide and protein results were filtered such that each contained less than 2% false positive identifications using a target decoy approach (*Elias and Gygi, 2007*). The list of proteins detected in each sample was used to construct a smaller database containing only these proteins, and the data were re-searched against this database using SEQUEST considering phosphorylation at serine, threonine, and tyrosine as variable modifications. Identifications were filtered to a false positive detection rate of no more that 5%. The discrete location of phosphorylation sites within each phosphopeptide detected were validated using the A-Score algorithm (*Beausoleil et al., 2006*).

### Imaging and analysis of Cdc24-mEOS, mEOS-Cdc42 and gic2$_{1-208}$-yEGFP

In order to visualise the Cdc24 phospho-mutant proteins in vivo, the DMY570 strain, containing Cdc24 under the control of a galactose-inducible promoter as the sole source of Cdc24 was transformed with pDM700, pDM701 and pDM704 plasmids, encoding Cdc24-mEOS, cdc24-46A-mEOS and cdc24-28D-mEOS, respectively. To visualise Cdc42 in the cdc24 phospho-mutant strains, DMY2147, DMY2151 and DMY2154 were transformed with pDM303, encoding mEOS-*CDC42*. To detect Cdc42 GTP in the cdc24 phospho-mutant strains, DMY2147, DMY2151, DMY2154, DMY2333 and DMY2334 were transformed with pDM842, encoding *gic2$_{1-208}$*-yEGFP. Transformants were plated on selective minimal media containing dextrose and were grown at 25°C for imaging in mid-logarithmic phase.

All Images were analysed and processed using ImageJ software. To calculate the enrichment of Cdc24, Cdc42 and gic2$_{1-208}$ at the pole, the average fluorescence intensity of the pole (AFIP) and the cytosol (AFIC) were determined for each cell using an empirically determined threshold value that enabled the cell pole to be identified. Next, the AFIP was normalised as follows: normalised fluorescence intensity of the pole= (AFIP-AFIC)/AFIC. The values were plotted using Prism software.

## Acknowledgements

We thank Jean-Louis Mergny for the use of the fluorescence plate reader and Lionel Minvieille-Sébastia for the anti-Fip1 antibody. We also thank Anne Royou, Doug Kellogg and Robert Arkowitz for comments on the manuscript. This work was supported by funding from ANR grant ANR-13-BSV2-0015-01, the Regional Council of Aquitaine, ARC, CNRS and the University of Bordeaux.

## Additional information

### Funding

| Funder | Grant reference number | Author |
|---|---|---|
| Centre National de la Recherche Scientifique | | Derek McCusker |
| Agence Nationale de la Recherche | ANR-13-BSV2-0015-01 | Derek McCusker |
| Regional Council of Aquitaine | 2012 13 01 012 | Derek McCusker |
| Regional Council of Aquitaine | 2015-1R30113 | Derek McCusker |

The funders had no role in study design, data collection and interpretation, or the decision to submit the work for publication.

### Author contributions

PR, Formal analysis, Investigation, Methodology, Writing—original draft; RM, Formal analysis, Investigation; CB, AM-L, CÜ, LB, Investigation; FSA, Methodology; SPG, Formal analysis, Supervision,

Methodology; DM, Conceptualization, Formal analysis, Supervision, Funding acquisition, Investigation, Writing—original draft, Writing—review and editing

**Author ORCIDs**
Caner Ünlü, http://orcid.org/0000-0002-0612-3111
Derek McCusker, http://orcid.org/0000-0003-1455-1711

---

## Additional files

**Supplementary files**
• Supplementary file 1. Yeast strains used in this study.

---

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
