## [Decision Letter]

[Editors’ note: a previous version of this study was rejected after peer review, but the authors submitted for reconsideration. The first decision letter after peer review is shown below.]

Thank you for submitting your work entitled "Scaffold-mediated gating of Cdc42 signaling flux contributes to GTPase nanoclustering" for consideration by *eLife*. Your article has been reviewed by three peer reviewers, and the evaluation has been overseen by Mohan Balasubramanian as Reviewing Editor and a Senior Editor. The reviewers have opted to remain anonymous.

Our decision has been reached after extensive consultation between the reviewers. Based on these discussions and the individual reviews below, we regret to inform you that your work will not be considered further for publication in *eLife*.

The referees agree that the reconstitution of Cdc42-Cdc24-Bem1, using Pichia expressed Cdc42, is well done and linking the phosphorylation status to self-limitation of Cdc42 activation (i.e. GEF activity of phosphorylated Cdc24 is not activated further by Bem1) is an exciting conclusion (with the proviso that there are a number of controls that need to be done). The genetic experiments and associated imaging with the phosphomimetic mutants are interesting, but need to be strengthened.

The problem is in Figures 6 and 7 (super-resolution microscopy) and this seems like a separate paper that has been appended to the rest of the work, which has technical and interpretational issues, all listed below.

In light of this, we will be unable to consider this paper further for publication in *eLife*. However, the referees believe that the in vitro reconstitution and the associated genetics could be submitted as a short-article (please take a look at *eLife* contribution types for more information). If you choose this approach, please address the substantive points raised by the referees pertaining to Figures 1-5 (most of which are control experiments /new experiments that should strengthen the findings). The referees’ comments are appended verbatim.

Reviewer #1:

In this manuscript, Rapali et al. study the regulation of an important complex that regulates cell polarization through Cdc42 GTPase. Cdc42 is activated by its GEF Cdc24, and also binds the scaffold Bem1. The authors show that Bem1 acts in vitro in two opposing ways: it promotes Cdc24 GEF activity towards Cdc42 and also increases the rate of phosphorylation of Cdc24 by the PAK kinase Cla4. Phosphorylated Cdc24 still activates Cdc42, but cannot be promoted by Bem1. This thus likely limits the positive effect that Bem1 has on Cdc42 activation, which the authors interpret as Bem1 contributing to signaling flux through Cdc42. They also identify all phosphorylation sites on Cdc24 and analyze the effects of phospho-mutants and phospho-mimetic mutants in vivo. In a second part, the authors use super-resolution imaging approaches to probe the diffusion and cluster-forming ability of Cdc42, probing both the function of Bem1 and of Cdc24 phosphorylation sites in these processes.

In my view, the first half of the manuscript is very interesting, clean and refreshing. The biochemistry is performed to high standards and largely convincing (a few missing controls are mentioned below). The interpretation that the contrasting roles of Bem1 increase flux is possible, but only one of several interpretations. For instance, another interpretation could be that Bem1 contributes to reaching rapidly a high, but not excessive Cdc42 activity. By contrast, the second half of the manuscript, including the analysis of Cdc24 phosphomutants and especially the super-resolution imaging, is less convincing and does not fit very well with the first half. It reads more like two stories lumped together. The consequence is that I do not feel the statement made by the title of the paper is well supported. Specific comments are below.

Regarding the effect of Cdc24 phosphorylation status on Bem1-dependent activity enhancement, one missing control in Figure 1 is to test the effect of 0.2 µM Bem1, which is the concentration remaining in the reaction in Figure 3. The expectation is that it should not have an effect at this concentration also in absence of Cla4-mediated phosphorylation. Another important missing control is to test whether the effect of Cla4 is solely through phosphorylation or also through protein binding. This should be addressed by using a kinase-dead allele in reaction as in Figure 3.

The interpretation of the Cdc24 phospho-mutants is complicated and difficult to link with the biochemistry shown in the first part of the paper. They appear to conclude from their in vivo analysis "in keeping with their in vitro biochemical observations" that the 46A is active and the 28D less active (subsection “Phospho-regulation of Cdc24 impacts polarized growth in vivo” and further). If this is "in keeping with the biochemical observations", the difference in activity in the two alleles should also be Bem1-dependent in vivo, not intrinsic to Cdc24 GEF activity (i.e., there should not be a difference between the two alleles in a bem1 mutant), but this is not tested. More generally, all effects of the *46A* mutant are interpreted relative to Bem1-Cla4 function. However, this mutant also includes the CDK sites, and thus some effects, such as delay in budding may be due to CDK1 action. It is also unclear in the text how the 28D sites were selected. The text could also sometimes be clearer. For instance, the double-negation in paragraph five of the section is convoluted. What the authors may mean is that the phospho-blocking and phospho-mimetic mutations have distinct phenotypes and thus are likely to affect Cdc24 each in a specific manner. Clarification on all these points would help the reader.

The mobility shift of Cdc24 from synchronized cultures is also difficult to interpret. The authors claim that "wild type Cdc24 undergoes a dramatic reduction in electrophoretic mobility due to phosphorylation at around 60 minutes after synchronous release from a G1-arrest". However, the Figure 4 shows very high level of phosphorylation at time 0, slightly coming down and then up again around 60min. The description of the result thus does not correspond to what is shown on the figure. This is likely due to the α-factor treatment use for synchronization, which may cause polarization responses. For the analysis here, it would likely be much better to synchronize cells by elutriation to probe this without interfering with cell polarization. Secondly, the authors show the reduced mobility is indeed due to phosphorylation by performing phosphatase treatment of wildtype and mutant Cdc24 alleles.

The effect of Cdc24 phosphorylation on Cdc42 distribution and activity in vivo are not clear. The authors interpret the enrichment of Cdc24 and Cdc42 at the polar cap as sign of Cdc42 activity. This could be more directly evaluated by using probes for Cdc42-GTP, such as CRIB, which would be predicted to be in increased amounts in the 46A mutant, and decreased in the 28D mutant. Pull-down of Cdc42-GTP could also be performed to probe this point.

Regarding the super-resolution analysis of Cdc42, one major question I have is which version of mEos is being used? Some versions have the tendency to dimerize, which would make fusion proteins prone to oligomerization. For instance, mEos2 has been shown to still dimerize, which raises questions about mEos, from which mEos2 is derived. mEos3.1 and 3.2 appear to be truly monomeric. Because the authors do not have a Cdc42 mutant that blocks cluster formation, it is difficult to be very conclusive about the validity of the observed nanoclusters. I also feel that the link between nanocluster size and Cdc42 activity status is unclear and not directly supported by evidence. The 46A and 28D mutants show changes in Cdc42 cluster sizes at cell poles, which the authors interpret as a link between Cdc42 activity and cluster size, but (as mentioned above) they have not directly tested Cdc42 activity in these mutants in vivo. To probe for a link between activity and cluster size, the authors should look at the distribution of constitutively active and inactive Cdc42 mutants.

I have similar difficulties with the analysis of diffusion coefficients. The differences between wt and bem1∆ are also very slight, which does not help to affirm the position that the observed nanoclusters are physiologically relevant. I find for instance surprising that, in Figure 6A-B, the diffusion coefficients seem to be altered in bem1∆ both at the pole and in the rest of the cell compared to wildtype (although statistics between bem1∆ and wt are not provided, so I am not sure this is statistically supported). I would expect a faster diffusion at the pole of bem1∆ cells, as shown (but again not statistically tested), but not a slower diffusion elsewhere. More generally, the analysis of Cdc42 trajectories is not particularly convincing. To convince that there are actually four distinct classes (as opposed to a continuum), the authors should show a plot of frequencies versus α value, which should reveal four distinct peaks/modes. From the examples shown, I am for instance not convinced that there is any directed movement: the examples show only two data points in a line before a more confined-looking behavior.

Reviewer #2:

In this manuscript the authors study the role of the scaffold protein Bem1 and its interactions with the Cdc42 GEF Cdc24 and the PAK kinase Cla4, in the context of budding yeast polarization. The first part of the manuscript focuses on biochemical experiments with purified proteins and shows that Bem1 increases GEF activity of Cdc24 on Cdc42 and at the same time supports phosphorylation of Cdc24 by the PAK kinase Cla4. Interestingly, the authors then find that phosphorylated Cdc24 can no longer be stimulated by Bem1, providing a possible self-limiting loop to Cdc42 activation – a critical feature for efficient cell polarization that has remained elusive on the molecular level in all existing models of spontaneous symmetry breaking. The authors then confirm the relevance of the identified reactions in vivo, using a Cdc24 mutant lacking 46 MS-mapped phosphorylation sites. In particular, using single particle tracking of Cdc42 in cells they propose that Bem1 and Cdc24-phosphorylation contribute to dynamics of Cdc42 by affecting nanocluster formation at the polarization site.

The topic of this study connects central aspect of signal transduction and cell polarization research. Bem1 has been proposed to be a key factor for a positive feedback loop driving symmetry breaking and the earliest steps in yeast polarization. However, the specific mechanisms and consequences of Bem1-Cdc24-Cla4 interactions have been subject to a long and controversial debate in the field. The current study clarifies a few important molecular aspects of this debate in a convincing manner, and provides elegant in vivo support for their main biochemical findings. I therefore think that this work is of great relevance and provides the kind of combination between conceptual advance and technical innovation aimed for at *eLife*. However, there are a couple of issues that definitely need to be clarified or improved before publication (see major points below). There are also a couple of technical concerns that can be easily fixed.

1) The demonstrations of Bem1 effects on Cdc24 activity and phosphorylation are convincing but they mostly confirm or incrementally improve on previous studies. The idea of a self-limitation and negative effect of phosphorylation on GEF-stimulation is however a really new and therefore very important part of this study. As I interpret the current results the negative effect on Bem1-stimulation could either be due to the phosphorylation itself (as the authors suggest) or due to the presence of Cla4 (which is always present in the relevant assays from Figure 3). Can the GEF assays in 3B/C be performed with a kinase dead version of Cla4 as control? Alternatively, the kinase reaction could be performed without phosphate in the medium. This would support the idea of a central effect of the Cdc24 phosphorylation sites. In the same direction: Bem1 seems to stimulate additional phosphorylation of very specific sites (not just more but different pattern). It would good to include a mutant where only the Bem1-specifi sites are mutated and include this in the assays for Figure 3. Finally, it should be possible to combine Dcla4 and Drga1 to show that this leads to a similar phenotype as the *46A* mutant + Dcla4 (this might have been already shown, in this case refer to the relevant paper).

2) The second key point of this study is the proposed effect of Bem1 on Cdc42 nanocluster formation. While I have no doubts that such clustering occurs for Cdc42 – as it probably does for every protein of the PM (Spira et al., NCB 2012) – the proposed link to Bem1 and phosphorylation is less convincing to me. My main concern stems from the single molecule localization analysis. The signal density on the cell surface (for both Cdc42 and Cdc24) seems very low, especially in the non-polarized regions. This could be due to the limitations of mEOS2 (permanent bleaching, low quantum yield) compared to chemical dyes used in STORM applications. How can the authors exclude that the cluster results are affected by labeling density (larger cluster in the denser polarized cap area)? This would perfectly correlate with the observed effects (increase at pole for 42A). An additional concern is the current resolution limit (40 nm), which is very close to the identified cluster size (76 nm). Considering additional factors such as mobility of the fluorophore (GAGA linker) and variable polarization of fluorescence (Shivanandan et al., FEBS letters 2014) clusters might not be readily resolvable any more. In summary, at the moment the shown PALM data definitely needs some sort of additional validation. Possible experimental controls with/without predicted clustering would be Cdc42 in LatA treated cells, constitutive active Cdc42 or markers of known clusters such a eisosomes (Pil1, Sur7) or endocytic patches (Ede1, Abp1).

3) A majority of the track in Figure 6 seem to occur at the edge of cells. If using HiLo (beter to use TRIRF) the focus should be on the actual cell surface so that all tracks can be followed in 2D. At the moment I see a bias to shorter tracks as the movement in z can no longer be tracked. Also the observed differences (pole vs. rest) are not very strong and the effect of Bem1 deletion seems to be rather increasing motility in non-polarized areas than decreasing the polar motility. To support the single particle tracking it would therefore be important to provide an additional approach. The authors could perform careful FRAP experiments in cap vs. non-polarized regions of the cell. This should allow detection of the observed effects. After all, the average MSD that was obtained is quite similar to the effective FRAP recovery rate.

4) The current data support a reduction in stimulation of Cdc24 by Bem1 after Cla4-mediated phosphorylation. To really support the sequence of events proposed in the model (Figure 8), Cla4 would need to act after Bem1 recruitment (fits the idea of Bem1 as recruitment platform for Cla4). Can the authors perform such an experiment with their reconstituted system? 1. Measure basal GEF activity, 2. Add Bem1 for 30 min, 3. Measure stimulated GEF activity, 4. Add Cla4 (high levels) for 1 h, 5. Measure GEF activity again to see basal levels again. This would provide a perfect final experiment and nicely relate to the proposed model.

Reviewer #3:

This manuscript investigates how the Cdc42 scaffold Bem1 regulates Cdc42 activation via positive and negative regulation in the budding yeast model system. Reports have shown that Cdc42 activation undergoes oscillations/fluctuations during polarization in the fission yeast (Das et al. 2012) and budding yeast (Howell et al., 2012) model systems. These findings indicated that Cdc42 undergoes both positive feedback as well as a time delayed negative feedback. Further Das et al., 2012 suggested that the negative feedback was mediated by the Cdc42 target Pak1 kinase and in Pak1 kinase mutants Cdc42 activation and GEF-scaffold complex (Scd1-Scd2) is increased at the cell tips leading to increased Cdc42 activation and dampening of Cdc42 pulsing at these sites. Later Kuo et al., 2014 demonstrated that Cdc24 is phosphorylated by the kinase Cla4 and this leads to a decrease in Cdc24 GEF activity. This manuscript provides further evidence that Cla4 phosphorylates Cdc24 and thus leading to dampening of GEF activity. The authors claim that Bem1 promotes Cla4 mediated phosphorylation of Cdc24 and describe this as a "Bem1 dependent mechanism". They also claim that this leads to desensitization of Cdc24 to Bem1 interaction. The evidence in support of these claims is not strong. Further, in support of previous reports, the authors use single particle tracking to show that the Cdc42 complex at the cell tips is less dynamic in the presence of Bem1. The authors suggest that Cdc24 phosphorylation decreases clustering of Cdc42 complexes at sites of polarization. While the findings presented here do not provide a novel insight into how Cdc42 is negatively regulated, it does provide additional evidence for previous findings.

1) The authors state that phosphorylation of Cdc24 reduces its activity, but do not directly test whether this is due to decreased affinity between Cdc24 and Cdc42. While the authors test the GEF activity of Cdc24 in various mutants, they fail to use the most direct (and biologically relevant) approach of measuring Cdc42 activity in vivo. Later they present phenotypes of these mutants, but only indirectly link them to Cdc42 by observing cell morphology and cytoskeleton organization (Figure 4).

2) Previous reports (Kozubowski et al., 2008) have shown that the Pak kinase forms a complex with Cdc42-Cdc24-Bem1 to positively regulate Cdc42 activation and constitutes a positive feedback loop that leads to symmetry breaking. The authors have not discussed their findings in the context of this report. Moreover, Cdc24-Cla4 fusion was sufficient to normally activate Cdc42 suggesting that Bem1 acts as a scaffold and may stabilize the Cdc42 ternary complex. This could explain increased Cdc24 phosphorylation in the presence of Bem1. Thus it is possible that Bem1 does not specifically influence Cla4 phosphorylation but rather any scaffold capable to bringing together Cdc42-Cdc24-Cla4 in a complex would work.

3) The authors include Cdk1 phosphorylation sites in their D and A cdc24 mutants, but do not describe whether these residues are phosphorylated during bud emergence, which would indicate their relevance. Inclusion of these residues in the S and A cdc24 mutants is problematic, because it prevents the changes in protein activity and localization to be solely attributed to the proposed Bem1-Cdc24-Cla4-dependent mechanism. Additionally, the authors do not provide rationale stating why only 28 residues were mutated in the D mutants, while 46 were used in the A mutants.

4) The authors do not mention time-delayed negative feedback regulation of Cdc42 as reported by Das et al. 2012 and Howell et al., 2012. Could their findings provide a molecular explanation for this feedback regulation? Kuo et al. 2014 have shown that Cdc24 non-phosphorylatable mutants show stabilized Cdc42 activation during bud emergence. Do the authors expect a similar phenotype with their *Cdc24-46A* mutants.

5) How does Cdc42 dynamics change with *cdc24-28D* and *46A* mutants? This would further explain the outcome of the Cdc24 phosphorylation on Cdc42 dynamics and polarization.

6) If Pak mediated Cdc24 phosphorylation does not alter GEF activity or Cdc24-Bem1 interaction what exactly does Cdc24 phosphorylation lead to? The authors claim that Cdc24 phosphorylation renders it insensitive to Bem1. However, they also show that Cdc24 non-phosphorylatable Cdc24 requires Bem1 for enhanced GEF activity. While these findings are not necessarily contradictory, they do lead to questions about how Cdc24 responds to Bem1 interaction. The authors claim that Cdc42 phosphorylation does not disrupt Cdc24-Bem1 interaction. Could it be that Cdc24 phosphorylation destabilizes that Cdc24-Cdc24-Bem1-Cla4 ternary complex instead?

[Editors’ note: what now follows is the decision letter after the authors submitted for further consideration.]

Thank you for resubmitting your work entitled "Scaffold-mediated gating of Cdc42 signaling flux" for further consideration at *eLife*. Your revised article has been favorably evaluated by Tony Hunter as the Senior editor, Mohan Balasubramanian as Reviewing editor, and three reviewers.

The referees have found your revised short format article significantly improved and are satisfied that you have provided a compelling biochemical reconstitution of Cdc42 GTPase signaling pathway, especially around the negative feedback regulation of the pathway and the role of Cla4 phosphorylation of Cdc24-GEF in a Bem1 dependent manner.

However, a single major and several minor points have been raised. I would like you to address the major point highlighted below, without fail. I will leave it to your judgement about the rest of the minor points, although several of these are very easily fixed with writing, doing which will improve the readability.

Essential revisions:

The authors have also added a more direct measure of Cdc42-GTP local levels by use of the Gic2 CRIB localization. They report that CRIB levels are higher in the Cdc24-A mutant and undetectable in the Cdc24-D mutants (Figure 3). This is a good addition, but I am not entirely convinced by this rather dramatic observation. Indeed, the CRIB signal is measured apparently in non-budded cells, but the authors report that the cdc24-D mutants have a delay in polarization (reported in Figure 2—figure supplement 1). What is the control that the cells that do not show a signal (28D and 15D) actually have a polarized patch of Cdc24? It would be important to quantify only cells that have actually polarized to test whether the local levels of active Cdc42 are indeed lower. One simple way would be to measure CRIB levels in small-budded cells. Another better way would be to co-image Cdc42-GFP and mCherry Gic2 CRIB. In Figure 3, the authors show that Cdc42 levels are unchanged in cdc24-D mutants, and so co-imaging Cdc42 and CRIB would be a powerful way of demonstrating that the localized Cdc42 is actually less active. If this is the case, this would be I think a first such instance of uncoupling Cdc42 localization and local activation.

---

## [Author Response]

[Editors’ note: the author responses to the first round of peer review follow.]

*Reviewer #1:*

*In this manuscript, Rapali et al. study the regulation of an important complex that regulates cell polarization through Cdc42 GTPase. Cdc42 is activated by its GEF Cdc24, and also binds the scaffold Bem1. The authors show that Bem1 acts in vitro in two opposing ways: it promotes Cdc24 GEF activity towards Cdc42 and also increases the rate of phosphorylation of Cdc24 by the PAK kinase Cla4. Phosphorylated Cdc24 still activates Cdc42, but cannot be promoted by Bem1. This thus likely limits the positive effect that Bem1 has on Cdc42 activation, which the authors interpret as Bem1 contributing to signaling flux through Cdc42. They also identify all phosphorylation sites on Cdc24 and analyze the effects of phospho-mutants and phospho-mimetic mutants in vivo. In a second part, the authors use super-resolution imaging approaches to probe the diffusion and cluster-forming ability of Cdc42, probing both the function of Bem1 and of Cdc24 phosphorylation sites in these processes.*

*In my view, the first half of the manuscript is very interesting, clean and refreshing. The biochemistry is performed to high standards and largely convincing (a few missing controls are mentioned below). The interpretation that the contrasting roles of Bem1 increase flux is possible, but only one of several interpretations. For instance, another interpretation could be that Bem1 contributes to reaching rapidly a high, but not excessive Cdc42 activity. By contrast, the second half of the manuscript, including the analysis of Cdc24 phosphomutants and especially the super-resolution imaging, is less convincing and does not fit very well with the first half. It reads more like two stories lumped together. The consequence is that I do not feel the statement made by the title of the paper is well supported. Specific comments are below.*

We thank the reviewer for their encouraging comments. The title of the manuscript has now been modified to reflect the focus on the biochemistry. We have also performed numerous additional experiments and controls to analyze the *cdc24* phospho-mutants in light of the comments raised during review. These are explained in detail below.

*Regarding the effect of Cdc24 phosphorylation status on Bem1-dependent activity enhancement, one missing control in Figure 1 is to test the effect of 0.2 µM Bem1, which is the concentration remaining in the reaction in Figure 3. The expectation is that it should not have an effect at this concentration also in absence of Cla4-mediated phosphorylation. Another important missing control is to test whether the effect of Cla4 is solely through phosphorylation or also through protein binding. This should be addressed by using a kinase-dead allele in reaction as in Figure 3.*

As requested, we performed GEF reactions with 0.2 µM Bem1 and have included the data in new Figure 1. We found a very modest stimulatory effect on GEF activity at this concentration. Additionally, we performed GEF assays to distinguish whether the effect of Cla4 on Bem1-stimulated GEF activity originates from phosphorylation or from protein binding. To do so, we performed GEF assays in the presence of Cla4, while omitting ATP. In new Figure 1, we report that Cla4 was unable to dampen Bem1 stimulation of GEF activity unless ATP was present, indicating that Cla4 exerts its effect on Cdc24 GEF activity via phosphorylation. These results are described in subsection “Phosphorylation of Cdc24 inhibits the scafford-dependent increase in Cdc24 GEF activity in vitro” of the resubmitted manuscript.

*The interpretation of the Cdc24 phospho-mutants is complicated and difficult to link with the biochemistry shown in the first part of the paper. They appear to conclude from their* in vivo *analysis "in keeping with their* in vitro *biochemical observations" that the 46A is active and the 28D less active (subsection “Phospho-regulation of Cdc24 impacts polarized growth* in vivo*” and further). If this is "in keeping with the biochemical observations", the difference in activity in the two alleles should also be Bem1-dependent* in vivo*, not intrinsic to Cdc24 GEF activity (i.e., there should not be a difference between the two alleles in a bem1 mutant), but this is not tested. More generally, all effects of the 46A mutant are interpreted relative to Bem1-Cla4 function. However, this mutant also includes the CDK sites, and thus some effects, such as delay in budding may be due to CDK1 action. It is also unclear in the text how the 28D sites were selected. The text could also sometimes be clearer. For instance, the double-negation in paragraph five of the section is convoluted. What the authors may mean is that the phospho-blocking and phospho-mimetic mutations have distinct phenotypes and thus are likely to affect Cdc24 each in a specific manner. Clarification on all these points would help the reader.*

We agree with the reviewer that our working model predicts that there should be no difference between the two *cdc24* alleles in a *bem1* mutant. In new Figure 2 we report that this is the case. The deletion of *BEM1* in *cdc24-46A* or *cdc24-28D* strains resulted in an indistinguishable temperature sensitive growth phenotype at 37ºC (Figure 2). In addition, we found identical morphological defects in *cdc24-46A bem1, cdc24-28D bem1* and *bem1* mutants (Figure 2).

In order to distinguish the influence of sites phosphorylated by Cla4 in the presence of Bem1 from all of the other sites in Cdc24 that are phosphorylated, we generated mutants in which only the Bem1-dependent sites were mutated to alanine or aspartate, which we refer to as *cdc24-15A* and *cdc24-15D* (Figure 2). The *cdc24-15D* mutant displayed a temperature sensitive growth defect (Figure 2) and the *cdc24-15D* mutant also displayed a delay in bud emergence in synchronized cultures (Figure 2—figure supplement 1). It therefore appears that the Bem1-dependent phosphorylation sites, rather than the Cdk1-dependent phosphorylation sites, contribute to the observed delay in bud emergence. This point is clarified in subsection “Phospho-regulation of Cdc24 impacts polarized growth in vivo” of the new submission.

The *cdc24-28D* mutant was generated after homologous recombination of a *cdc24-46D* construct into the *CDC24* locus. This is now explained in subsection “Plasmid construction”. It seems likely that the *cdc24-46D* mutant was not recovered after transformation because the mutant is not viable.

We apologize for the unclear text and we are grateful for the alternative that you propose. The text referred to has now been modified. In subsection “Phospho- regulation of Cdc24 impacts polarized growth in vivo” paragraph five now read, "These results indicate that the phospho-blocking and phospho-mimetic mutations confer distinct phenotypes and are thus likely to affect Cdc24 in a specific manner". We have re-read the text to try and find other instances of unclear text, but please let us know if we can help clarify any additional points.

*The mobility shift of Cdc24 from synchronized cultures is also difficult to interpret. The authors claim that "wild type Cdc24 undergoes a dramatic reduction in electrophoretic mobility due to phosphorylation at around 60 minutes after synchronous release from a G1-arrest". However, the Figure 4 shows very high level of phosphorylation at time 0, slightly coming down and then up again around 60min. The description of the result thus does not correspond to what is shown on the figure. This is likely due to the α-factor treatment use for synchronization, which may cause polarization responses. For the analysis here, it would likely be much better to synchronize cells by elutriation to probe this without interfering with cell polarization. Secondly, the authors show the reduced mobility is indeed due to phosphorylation by performing phosphatase treatment of wildtype and mutant Cdc24 alleles.*

We apologize to the reviewer for our imprecise description of the text on this point. We have now edited the text of the new submission to more accurately describe the data. We agree with the reviewer that elutriation would be the optimal method of showing the cell cycle-dependent change in electrophoretic mobility of Cdc24. However, having searched extensively, there are no centrifugal elutriators at the present time in Bordeaux. Moreover, the cell cycle-dependent phosphorylation of Cdc24 has previously been demonstrated in wild type cells (see PMID 17417630 Figure 3 and PMID 11106754 Figure 4). The main point that we are drawing the reader's attention to in new Figure 2 is the loss of cell cycle-dependent Cdc24 phosphorylation in the *cdc24-46A* and *cdc24-28D* mutants, indicating that the phosphorylation sites identified by our mass spectrometry analyses are relevant in vivo. For the last point, the reviewer is correct that >90% Cdc24 migrates as a single species after phosphatase treatment. See, for example, PMID 24631237 Figure 1 and Figure 2.

*The effect of Cdc24 phosphorylation on Cdc42 distribution and activity* in vivo *are not clear. The authors interpret the enrichment of Cdc24 and Cdc42 at the polar cap as sign of Cdc42 activity. This could be more directly evaluated by using probes for Cdc42-GTP, such as CRIB, which would be predicted to be in increased amounts in the 46A mutant, and decreased in the 28D mutant. Pull-down of Cdc42-GTP could also be performed to probe this point.*

In order to clarify the role of Cdc24 phosphorylation on Cdc42 distribution and activity, we followed the reviewer's suggestion and performed quantitative image analyses of Cdc42-GTP using the CRIB domain encoded within Gic2. In new Figure 3, we find that this reporter of Cdc42-GTP is enriched at the pole of *cdc24-46A* cells compared to wild type cells, while it was undetectable at the pole of *cdc24-28D* cells. These results are described in subsection “Phospho- regulation of Cdc24 impacts polarized growth in vivo.” of the new submission. Thus, Cdc24, Cdc42 and a marker for Cdc42-GTP show enrichment in *cdc24-46A* cells and a diminution in *cdc24-28D* cells in vivo. We also repeatedly attempted to perform pull down assays of Cdc42-GTP, but the experiment did not work for reasons that we do not currently understand.

*Reviewer #2:*

*[…] 1) The demonstrations of Bem1 effects on Cdc24 activity and phosphorylation are convincing but they mostly confirm or incrementally improve on previous studies. The idea of a self-limitation and negative effect of phosphorylation on GEF-stimulation is however a really new and therefore very important part of this study. As I interpret the current results the negative effect on Bem1-stimulation could either be due to the phosphorylation itself (as the authors suggest) or due to the presence of Cla4 (which is always present in the relevant assays from Figure 3). Can the GEF assays in 3B/C be performed with a kinase dead version of Cla4 as control? Alternatively, the kinase reaction could be performed without phosphate in the medium. This would support the idea of a central effect of the Cdc24 phosphorylation sites. In the same direction: Bem1 seems to stimulate additional phosphorylation of very specific sites (not just more but different pattern). It would good to include a mutant where only the Bem1-specifi sites are mutated and include this in the assays for Figure 3. Finally, it should be possible to combine Dcla4 and Drga1 to show that this leads to a similar phenotype as the 46A mutant + Dcla4 (this might have been already shown, in this case refer to the relevant paper).*

We performed additional experiments that are now presented in Figure 1 of the new submission, in which we performed GEF assays with Cla4 and Bem1, but lacking ATP. The results indicate that Cla4 limits GEF activity via phosphorylation rather than via a protein-protein interaction, since Cdc24 GEF activity was stimulated by reactions containing Bem1 and Cla4 when ATP was omitted. However, when ATP was added to the GEF reactions, Cla4 attenuated Bem1 stimulation. The experiment is described in subsection “Phosphorylation of Cdc24 inhibits the scaffold-dependent increase in 135 Cdc24 GEF activity in vitro.”.

We agree with the reviewer that in addition to stimulating the rate of Cdc24 phosphorylation, Bem1 also increases the extent of Cdc24 phosphorylation by promoting the phosphorylation of sites that Cla4 alone does not seem to phosphorylate, even after extensive incubation with Cdc24 in kinase reactions. These results were also borne out by our mass spectrometry analysis. We have therefore clarified this point in the sub-heading “Bem1 increases the rate and extent of Cdc24 phosphorylation by the 116 PAK Cla4” and in the text.

We followed the reviewer's suggestion of generating mutants in which we mutated the Bem1-specific phosphorylation sites in Cdc24 to alanine or aspartate. We refer to these mutants as *cdc24-15A* and *cdc24-15D*. We found that, as with the *cdc24-28D* mutant, the *cdc24-15A* mutant was temperature sensitive. This data is shown in Figure 2 of the revised submission. We also looked at the levels of Cdc42-GTP in vivoin these mutants. The *cdc24-15A* mutant showed a higher level of the CRIB construct compared to wild type *CDC24* at the pole, but less than the *cdc24-46A* mutant. In the *cdc24-15D* mutant, levels of the CRIB construct were below the levels required for reliable quantification. The results are presented in new Figure 3 and explained in subsection “Phospho- regulation of Cdc24 impacts polarized growth in vivo”.

The phenotype of the *cla4 rga1* double mutant has previously been reported as being elongated in a similar manner to the *cdc24-46A rga1* double mutant. Thus, the data that we are reporting are complementary with previous studies. We have now cited this work, Caviston et al., 2003.

*[…] 4) The current data support a reduction in stimulation of Cdc24 by Bem1 after Cla4-mediated phosphorylation. To really support the sequence of events proposed in the model (Figure 8), Cla4 would need to act after Bem1 recruitment (fits the idea of Bem1 as recruitment platform for Cla4). Can the authors perform such an experiment with their reconstituted system? 1. Measure basal GEF activity, 2. Add Bem1 for 30 min, 3. Measure stimulated GEF activity, 4. Add Cla4 (high levels) for 1 h, 5. Measure GEF activity again to see basal levels again. This would provide a perfect final experiment and nicely relate to the proposed model.*

We agree that this is a great test of the model, but it has been extremely technically challenging. We perform the assays in 10 µl volumes, so the extended times required to extract the Kobs from the reaction curves meant that the reactions become prone to evaporation. The problems are not remedied by increasing the reaction volumes, since the next problem encountered is judging the effect of phosphorylation. The effect of adding Bem1 to the GEF activity is clear, but the effect of adding kinase is to subtly dampen the rate of GEF activity. There are additional problems here: most of the substrate, Cdc42, has been consumed early in the reaction, resulting in increased signal-to-noise as the reaction proceeds. Moreover, the reaction rate can no longer be fit to a single exponential, but must instead be judged manually. In conclusion, we cannot currently do this experiment, and in fact, it was for this reason that we initially broke this complex reaction into more approachable constituents.

*Reviewer #3:*

*[…] 1) The authors state that phosphorylation of Cdc24 reduces its activity, but do not directly test whether this is due to decreased affinity between Cdc24 and Cdc42. While the authors test the GEF activity of Cdc24 in various mutants, they fail to use the most direct (and biologically relevant) approach of measuring Cdc42 activity* in vivo*. Later they present phenotypes of these mutants, but only indirectly link them to Cdc42 by observing cell morphology and cytoskeleton organization (Figure 4).*

We have now analyzed a marker for active Cdc42 in vivo, as requested by the reviewer. The results are presented in new Figure 3 and described in subsection “Phospho- regulation of Cdc24 impacts polarized growth in vivo”. We find that the marker of active Cdc42 is enriched at the pole of the *cdc24- 46A* mutant compared to wild type *CDC24*, while the levels of active Cdc42 in the *cdc24-28D* mutant were below the detection limit of our imaging system. Thus, the in vitroGEF assays, actin cytoskeleton staining, Cdc24 localization, Cdc42 localization and active Cdc42 localization are consistent.

*2) Previous reports (Kozubowski et al. 2008) have shown that the Pak kinase forms a complex with Cdc42-Cdc24-Bem1 to positively regulate Cdc42 activation and constitutes a positive feedback loop that leads to symmetry breaking. The authors have not discussed their findings in the context of this report. Moreover, Cdc24-Cla4 fusion was sufficient to normally activate Cdc42 suggesting that Bem1 acts as a scaffold and may stabilize the Cdc42 ternary complex. This could explain increased Cdc24 phosphorylation in the presence of Bem1. Thus it is possible that Bem1 does not specifically influence Cla4 phosphorylation but rather any scaffold capable to bringing together Cdc42-Cdc24-Cla4 in a complex would work.*

The reviewer raises the complexity of the budding yeast polarity system, which has generated considerable debate in the field (see the comment published by Li & Wedlich-Soldner in the same issue of Current Biology as the Kozubowski paper, PMID 19278629). As the reviewer rightly points out, in the Kozubowski manuscript, the role of Bem1 in symmetry breaking was attributed to its tethering of Cdc24 and Cla4, since a Cdc24-Cla4 fusion bypassed the requirement for Bem1 and Bud1 in polarity establishment. This argued that Cla4 plays a positive role in polarity establishment. However, the role of Cla4 has since been shown to be more complex, as the Lew lab have published two manuscripts in which Cla4 was found to contribute to negative feedback during polarity establishment (PMID 24631237 and 26523396). The two findings are not incompatible; it is possible that Cla4 phosphorylates a subset of Cdc24 sites that boost Bem1-dependent GEF activity, but that the ensuing phosphorylation of all sites attenuates this stimulation. A similar mechanism has been reported for the Cdk1 inhibitor Wee1 in budding yeast (PMID 16096060). The initial phosphorylation of Wee1 by Cdk1 activates Wee1, while subsequent full phosphorylation of Wee1 by Cdk1 is inhibitory. It is not possible for us to perform the exhaustive mutational analyses required to test this hypothesis in the short article format of our resubmission, but we have cited the Kozubowski paper, as requested. In the Discussion section, we have now added that, "While a previous study found that the PAK Cla4 played a positive role in polarity establishment in S. cerevisiae (Kozubowski et al., 2008), subsequent work found that phosphorylation of Cdc24 by Cla4 resulted in reduced GEF activity, while increased Cdc42 recruitment to the cell pole was observed in Cdc24 phosphorylation mutants (Kuo et al., 2014; Wu et al., 2015)."

The reviewer asks if any scaffold that tethers Cdc42-Cdc24-Cla4 might suffice to stimulate Cdc24 phosphorylation. We have not tested whether other scaffolds might induce the same pattern of phosphorylation in Cdc24 as Bem1 and we have stated this in the Discussion section of the resubmitted manuscript: "It will also be important to test whether any scaffold that increases the local Cdc24-Cla4 concentration would stimulate Cdc24 phosphorylation, or whether Bem1 exerts specific effects." While Bem1 presumably does increase the local Cdc24 concentration available for phosphorylation by Cla4, our data suggest that it does so in a specific fashion. Even when Cla4 and Cdc24 are incubated together for extensive periods (6 hours), the extent of Cdc24 phosphorylation is markedly reduced compared with the inclusion of Bem1 in the reaction. Since no other proteins are present in these reactions, it seems likely that such extensive incubations in the absence of Bem1 would eventually result in full phosphorylation of Cdc24, but it does not. This point was also borne out by our mass spectrometry analysis in which we identified Bem1-specific phosphorylation sites in Ccd24.

*3) The authors include Cdk1 phosphorylation sites in their D and A cdc24 mutants, but do not describe whether these residues are phosphorylated during bud emergence, which would indicate their relevance. Inclusion of these residues in the S and A cdc24 mutants is problematic, because it prevents the changes in protein activity and localization to be solely attributed to the proposed Bem1-Cdc24-Cla4-dependent mechanism. Additionally, the authors do not provide rationale stating why only 28 residues were mutated in the D mutants, while 46 were used in the A mutants.*

In our resubmitted manuscript we generated mutants in which only Bem1- dependent sites in Cdc24 were mutated to alanine or aspartate. We refer to these mutants as *cdc24-15A* and *cdc24-15D*. These mutants resulted in enrichment of Cdc42-GTP at the pole in the case of the *cdc24-15A* mutant (Figure 3) and a temperature sensitive phenotype in the case of the *cdc24-15D* mutant (Figure 2). These mutants do not contain the Cdk1-dependent sites that we previously identified.

The *cdc24-28D* mutant was generated by homologous recombination after transformation of a *cdc24-46D* construct into yeast. Our interpretation is that mutation of all 46 phosphorylated residues to aspartate is inviable, whereas the *cdc24-28D* mutant supports viability, albeit with temperature sensitivity. We have explained this more clearly in subsection “Plasmid construction” of the Materials and methods section in the resubmission.

*4) The authors do not mention time-delayed negative feedback regulation of Cdc42 as reported by Das et al. 2012 and Howell et al., 2012. Could their findings provide a molecular explanation for this feedback regulation? Kuo et al. 2014 have shown that Cdc24 non-phosphorylatable mutants show stabilized Cdc42 activation during bud emergence. Do the authors expect a similar phenotype with their Cdc24-46A mutants.*

This is a very good point and is something that we are currently investigating using standard widefield imaging and SPT-PALM. The reviewer is correct that the current models from the Verde and Lew labs would predict that Cdc42 might be stabilized in the *cdc24-46A* mutant.

*5) How does Cdc42 dynamics change with cdc24-28D and 46A mutants? This would further explain the outcome of the Cdc24 phosphorylation on Cdc42 dynamics and polarization.*

We have only looked at Cdc42 dynamics in the mutants by single particle tracking. Cdc42 diffusion was reduced in the *cdc24-46A* mutant, both at the pole and non-pole, but this data has been removed from the resubmitted manuscript at the request of the editor.

6) If Pak mediated Cdc24 phosphorylation does not alter GEF activity or Cdc24-Bem1 interaction what exactly does Cdc24 phosphorylation lead to? The authors claim that Cdc24 phosphorylation renders it insensitive to Bem1. However, they also show that Cdc24 non-phosphorylatable Cdc24 requires Bem1 for enhanced GEF activity. While these findings are not necessarily contradictory, they do lead to questions about how Cdc24 responds to Bem1 interaction. The authors claim that Cdc42 phosphorylation does not disrupt Cdc24-Bem1 interaction. Could it be that Cdc24 phosphorylation destabilizes that Cdc24-Cdc24-Bem1-Cla4 ternary complex instead?

We apologize for giving the wrong impression on this point. The pulldown experiment that we present only allows us to conclude that phosphorylation of Cdc24 does not markedly influence its affinity for Bem1. Since the sensitivity of a pulldown assay is limited, further biophysical investigation will be required to fully address this point. It is possible that phosphorylation may increase the off-rate of Cdc24 for Bem1 or decrease the on-rate, but our results do not currently enable definitive conclusions on this point. Similarly, we are looking at how phosphorylation may affect the ternary Cdc42-Cdc24-Bem1-Cla4 complex. However, the labile nature of the complex, involving low affinity interactions between most components, makes this a very challenging project.

[Editors' note: the author responses to the re-review follow.]

*Essential revisions:*

*The authors have also added a more direct measure of Cdc42-GTP local levels by use of the Gic2 CRIB localization. They report that CRIB levels are higher in the Cdc24-A mutant and undetectable in the Cdc24-D mutants (Figure 3). This is a good addition, but I am not entirely convinced by this rather dramatic observation. Indeed, the CRIB signal is measured apparently in non-budded cells, but the authors report that the cdc24-D mutants have a delay in polarization (reported in Figure 2—figure supplement 1). What is the control that the cells that do not show a signal (28D and 15D) actually have a polarized patch of Cdc24? It would be important to quantify only cells that have actually polarized to test whether the local levels of active Cdc42 are indeed lower. One simple way would be to measure CRIB levels in small-budded cells. Another better way would be to co-image Cdc42-GFP and mCherry Gic2 CRIB. In Figure 3, the authors show that Cdc42 levels are unchanged in cdc24-D mutants, and so co-imaging Cdc42 and CRIB would be a powerful way of demonstrating that the localized Cdc42 is actually less active. If this is the case, this would be I think a first such instance of uncoupling Cdc42 localization and local activation.*

We redesigned the marker used to assay levels of Cdc42-GTP in vivoand we repeated the experiments shown in Figure 3. By extending the reporter construct that recognises Cdc42-GTP from residues 1-192 to 1-208 of Gic2, changing the promoter and swapping the tag from the N- to the C-terminus of Gic2, we found that the signal from the reporter was brighter, enabling us to monitor Cdc42-GTP levels in all of the *cdc24-Ala* and *cdc24-Asp* phosphorylation site mutants. Importantly, the optimization of this Cdc42-GTP reporter enabled us to ensure that quantitative imaging was only performed on polarizing cells, as requested by the reviewers. The conclusion of these experiments is unchanged from our previous submission, but we are now able to present corroborating quantitative analyses for the *cdc24-28D* and *cdc24- 15D* mutants.